# A model for focal seizure onset, propagation, evolution, and progression

Jyun-you Liou[1,2,3]*, Elliot H Smith[4†], Lisa M Bateman[3], Samuel L Bruce[5], Guy M McKhann[4], Robert R Goodman[4‡], Ronald G Emerson[3§], Catherine A Schevon[3], LF Abbott[1,6]

[1]Department of Physiology and Cellular Biophysics, Columbia University, New York, United States; [2]Department of Anesthesiology, NewYork-Presbyterian Hospital/Weill Cornell Medicine, New York, United States; [3]Department of Neurology, Columbia University Medical Center, New York, United States; [4]Department of Neurological Surgery, Columbia University Medical Center, New York, United States; [5]Vagelos College of Physicians & Surgeons, Columbia University, New York, United States; [6]Mortimer B. Zuckerman Mind Brain Behavior Institute, Department of Neuroscience, Columbia University, New York, United States

*For correspondence:
jyl9010@nyp.org

Present address: †Department of Neurosurgery, University of Utah, Salt Lake City, United States; ‡Department of Neurosurgery, Donald and Barbara Zucker School of Medicine at Hofstra/Northwell, Hempstead, United States; §Hospital for Special Surgery, Weill Cornell Medical College, New York, United States

Competing interests: The authors declare that no competing interests exist.

**Abstract** We developed a neural network model that can account for major elements common to human focal seizures. These include the tonic-clonic transition, slow advance of clinical semiology and corresponding seizure territory expansion, widespread EEG synchronization, and slowing of the ictal rhythm as the seizure approaches termination. These were reproduced by incorporating usage-dependent exhaustion of inhibition in an adaptive neural network that receives global feedback inhibition in addition to local recurrent projections. Our model proposes mechanisms that may underline common EEG seizure onset patterns and status epilepticus, and postulates a role for synaptic plasticity in the emergence of epileptic foci. Complex patterns of seizure activity and bi-stable seizure end-points arise when stochastic noise is included. With the rapid advancement of clinical and experimental tools, we believe that this model can provide a roadmap and potentially an in silico testbed for future explorations of seizure mechanisms and clinical therapies.

## Introduction

Focal seizures have been recognized for more than 3000 years, with descriptions dating back to ancient Mesopotamia (*Worthington, 2005*). Although focal seizures can present with a plethora of behavioral manifestations that vary according to the affected cortical regions, there are several consistent clinical and large-scale EEG features (*Kotagal et al., 2008*): propagation from a focal onset location to large brain regions, widespread neuronal synchronization, a transition from tonic to clonic activity, and a slowing pace of neuronal discharging prior to simultaneous seizure termination.

Our recent investigations, utilizing microelectrode array recordings in humans, identified neuronal underpinnings of these common seizure features (*Schevon et al., 2012*; *Smith et al., 2016*). Based on these findings and results from animal model studies (*Trevelyan et al., 2006*; *Trevelyan et al., 2007a*; *Trevelyan et al., 2007b*; *Wenzel et al., 2017*; *Wenzel et al., 2019*), we proposed a dual spatial structure for focal seizures consisting of a core region of seizing brain bounded by an ictal wavefront surrounded by a passively reactive penumbra. What delineates the boundary of the ictal core is the ictal wavefront, a narrow band of intense, desynchronized multiunit (tonic) firing that marks the transition to seizure at a given brain location. Typically, this tonic firing structure is not detectable in clinical EEG or band-limited local field potentials (LFPs). Evidence from human (*Schevon et al., 2012*) and animal studies (*Trevelyan et al., 2006*; *Trevelyan et al., 2007b*) suggests that collapse of inhibition is the key element causing ictal wavefront propagation, which leads

to progressive seizure territory expansion. The slow pace of ictal wavefront propagation (<1 mm/ sec) corresponds to the slow evolution of the electrographic seizure and clinical semiology, for example the classic Jacksonian march (*York and Steinberg, 2011*). As it advances, the ictal wavefront generates fast-moving ictal discharges (*Trevelyan, 2009*; *Smith et al., 2016*), directed inward towards the initiation point, at speeds two orders of magnitude higher than the wavefront propagation (*Smith et al., 2016*; *Liou et al., 2017*). These fast-moving ictal discharges are the basis of the well-recognized high amplitude field potential deflections that are the hallmark of electrographic seizures. Thus, there are two types of moving waves that characterize seizures: fast, inward-moving ictal discharges and the slow outward-moving wavefront of ictal recruitment.

To date, despite extensive prior work in computational seizure modeling (*Soltesz and Staley, 2011*), no theoretical study has shown how this dynamic topological structure can arise from basic biophysical mechanisms. Phenomenological models, such as Epileptor (*Jirsa et al., 2014*; *Proix et al., 2018*), have successfully utilized large-scale functional EEG features to reproduce the large-area, apparently synchronized EEG activity that characterizes clinical seizure recordings, although identifying smaller scale neurophysiological processes corresponding to each abstract variable in these models can be challenging. Biophysical models have been used to explore the role of specific neurophysiological processes responsible for seizure transitions, such as intracellular and extracellular potassium dynamics (*Cressman et al., 2009*; *Ullah et al., 2009*), transmembrane chloride gradients (*Buchin et al., 2016*), and calcium-activated processes (*Yang et al., 2005*). However, the evolving, wide-area dynamics that characterize the life cycle of a seizure remain difficult to explain by any single biophysical mechanism. Moreover, given that seizures can simultaneously activate a multitude of pathophysiological processes across extensive interconnected brain regions, biophysical models may be limited to a fragmented or overly specific account of seizure dynamics.

Here, we describe a biophysically-constrained cortical network model designed to link the key pathological cellular mechanisms that underpin seizures to their large-scale spatial structure. Inspired by the spirit of phenomenological models, we adopt an approach with minimal assumptions, aiming to show that complex spatiotemporal dynamics can arise from simple, generalizable, and experimentally validated biophysical principles. Our modeling philosophy is to eschew model features that are inessential or that do not contribute directly or dominantly to the phenomena being considered. This allows us to identify basic mechanisms. In addition, while we, of course, build specific mechanisms into our models, our aim is to highlight the underlying biophysical properties that lead to pathology, so that the lessons learned are more general than the specific models. Using a targeted parameter search, we demonstrate that maintaining the normal transmembrane chloride gradient is critical for inhibition robustness, which is necessary for restricting seizure propagation. Our theory provides a theoretical framework explaining the key clinical features widely observed in focal seizure patients (*Kotagal et al., 2008*; *Ebersole et al., 2014*; *Extercatte et al., 2015*). We also test several predictions arising from the model using our existing dataset of microelectrode recordings of spontaneous human seizures.

## Results

We modeled the neocortex as a 2-dimensional neuron sheet (*Bressloff, 2014*). Model neurons are pyramidal cell-like and contain two intrinsic conductances – leak and slow potassium – and two synaptic conductances – excitatory (AMPA) and inhibitory (GABA-A). Neuronal mean firing rates are calculated by passing the difference between membrane potentials and thresholds through a sigmoid function. Model neurons incorporate spike-frequency adaptation mechanisms – neuronal firing increases the spike threshold and activates an additional slow afterhyperpolarization (sAHP) conductance. Model neurons are recurrently connected by direct excitatory projections (*Figure 1A*). They also inhibit each other indirectly through di-synaptic pathways via interneurons, whose dynamics are simplified in this study (see Materials and methods for more information regarding interneuron simplification). The effects of recurrent projections between model neurons are hypothesized to be distance-dependent, with the range of di-synaptic recurrent inhibition longer than the mono-synaptic excitatory range, thereby making the spatial distribution of the effective synaptic weights from a model neuron follows a 'Mexican hat' structure (*Prince and Wilder, 1967*; *Coombes, 2005*; *Bressloff, 2014*). In addition, our model includes a distance-independent recurrent inhibition pathway (*Figure 1A*, part γ), inspired by recent observation of large-scale inhibitory effects of focal seizure

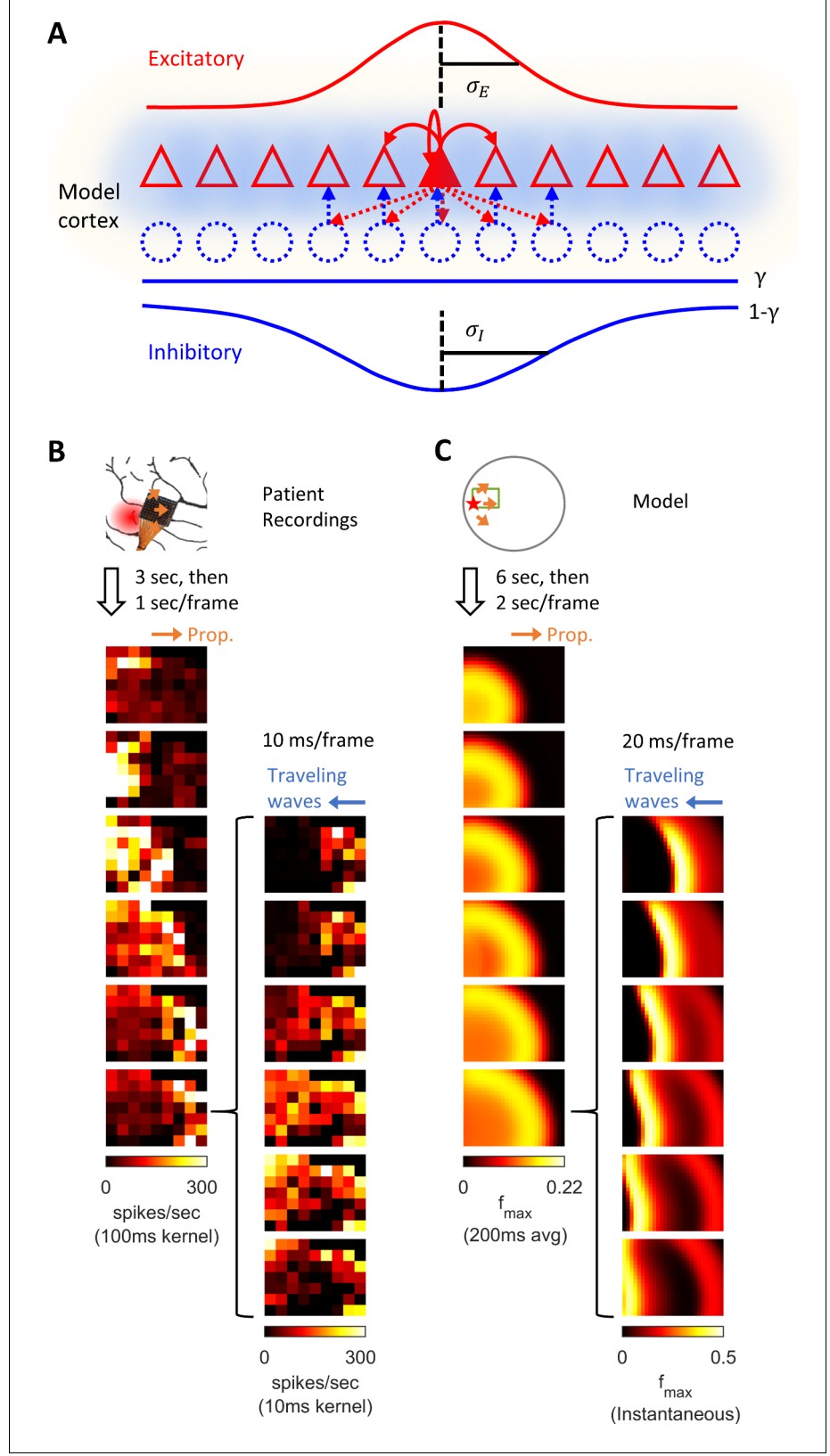

**Figure 1.** Model schematics and the comparison of patient recordings with model simulation results. (**A**) Model schematic. Triangles: model neurons. Red solid arrows: excitatory recurrents with distance-dependent strength of spatial kernel width $\sigma_E$. Dashed blue circles: inhibitory neurons. Dashed red-to-blue arrows: distance-dependent

*Figure 1 continued on next page*

*Figure 1 continued*

di-synaptic recurrent inhibition with spatial kernel width $\sigma_I$, which accounts for $1 - \gamma$ of the total recurrent inhibition. The remaining fraction $\gamma$ of the inhibition is distance independent and represented by the blue hues around model neurons. Interneuron membrane potentials are not explicitly modelled. (**B**) Microelectrode array recordings from Patient A. The red shaded area in the cartoon panel represents the clinically identified seizure onset zone. Orange arrows indicate the seizure propagation direction. The left column shows the spatiotemporal dynamics of multiunit activity in a slow timescale. The right column is a fast timescale zoom-in of the left bottom panel. Evolution of multiunit activity in a slow/fast timescale is estimated by convolving the multiunit spike trains with a 100/10 ms Gaussian kernel respectively. Left-to-right seizure recruitment was seen 3 s after the seizure onset (the orange arrow). Once recruited (left bottom panel), fine temporal resolution panels (the right column) showed right-to-left fast waves arose at the edge of the seizure territory (the blue arrow) and traveled toward the internal domain (fast inward traveling waves). (**C**) 2D rate model simulation results. Figure conventions adopted from B. The red star indicates where the seizure-initiating external current input was given ($I_d$ = 200 pA, duration 3 s, covering a round area with radius 5% of the whole neural sheet). Evolution of the neuronal activities within the green rectangle are shown in the panels. Notice the slow outward advancement of the seizure territory and the fast-inward traveling waves.

The online version of this article includes the following video for figure 1:

**Figure 1—video 1.** Full evolution of the model seizure shown in *Figure 1C*.

https://elifesciences.org/articles/50927#fig1video1

---

activity (*Eissa et al., 2017*; *Liou et al., 2018*). Biophysical parameters are set based on previously reported values from recorded neocortical pyramidal neurons (*Table 1*). In its original form, the model does not spontaneously seize. Instead model seizures are initiated by transient focal excitatory inputs. This model can be considered a model of a healthy neural circuit, with the degree of resistance to seizure controlled by its parameters.

Throughout the manuscript, variations of the primary 2D rate model are introduced to provide more in-depth assessment of key features. Simulation results of the primary model in a noise-free environment are first presented and qualitatively compared to patient recordings (*Figure 1*). Next, the model is simplified to 1-dimensional space to better illustrate physiological mechanisms that account for the slow advancement of the ictal wavefront and the generation of fast inward traveling waves (*Figures 2* and *3*). A corresponding 1-dimensional spiking model is subsequently shown in *Figures 4* and *5* to validate our rate-based approach and explore the effects of spike-timing dependent plasticity on spontaneous seizure occurrence. Finally, the main model is revisited with the addition of background current noise to explain bi-stable seizure evolution endpoints and the origin of complex spatial configurations of focal seizure activities (*Figures 6* and *7*). While such variations of the primary model are used to more clearly illustrate key dynamical features, the dynamics presented here are common across model variations.

## Comparison between patient microelectrode recordings and model simulation results

*Figure 1B* shows a 96-channel Utah microelectrode array recording from a patient experiencing a typical spontaneous neocortical-onset seizure (*Source data 1*). The human recording demonstrates a slow, progressive advancement of seizing neuronal activity from the left to right side of the array-sampled area (*Figure 1B*, left, orange arrows). This seizure recruitment was led by the ictal wavefront, which passed through the microelectrode-sampled brain region over the course of a few seconds. Following the brief period of tonic firing marking the passage of the ictal wavefront, the neurons, now inside the ictal core, transitioned to repetitive bursting (*Figure 1B*, right). Bursts propagated sequentially in space from the ictal wavefront back toward the internal domain, constituting fast-moving 'inward traveling waves' (*Figure 1B*, right, the blue arrow). Figure panel 1C shows similar activity patterns generated by our model. In the model, a self-sustaining tonic-firing region was established by a transient focal external stimulus (the red star in *Figure 1C*). The key dynamics seen in the human recordings were reproduced by the model, including the slow-marching wavefront (*Figure 1C*, right, orange arrows) and the subsequent fast inward traveling waves (*Figure 1C*, right, the blue arrow). *Figure 1—video 1* shows the full spatiotemporal evolution of this model seizure, and data from the Utah microelectrode arrays can be found online (*Source data 1*).

**Table 1.** Rate model parameters.

Neurons are modelled analogous to neocortical pyramidal neurons in terms of cell capacitance, leak conductance, and membrane time constant (*Tripathy et al., 2014*). Maximal synaptic conductances during seizures have been reported in the range of a few hundreds of nS (*Neckelmann et al., 2000*). Chloride clearance (*Deisz et al., 2011*), buffer (*Marchetti, 2005*), and seizure-induced sAHP conductance (*Alger and Nicoll, 1980*) are modeled according to the reported ranges. Spatially non-localized recurrent projections ($\gamma$) are considered in this study (*Liou et al., 2018*). All rate models are based on this standard parameter set. Any further parameter adjustment is reported in figure legends.

| Parameters | | unit |
|---|---|---|
| $C$ | 100 | pF |
| $g_L$ | 4 | nS |
| $\bar{g_E}$ | 100 | nS |
| $\bar{g_I}$ | 300 | nS |
| $E_L$ | -58 | mV |
| $E_E$ | 0 | mV |
| $E_K$ | -90 | mV |
| $f_{max}$ | 200 | Hz |
| $\beta$ | 2.5 | mV |
| $\tau_E$ | 15 | ms |
| $\tau_I$ | 15 | ms |
| $\tau_\phi$ | 100 | ms |
| $\phi_0$ | -45 | mV |
| $\Delta_\phi$ | 0.3 | mV/Hz |
| $\tau_{Cl}$ | 5 | second |
| $V_d$ | 0.24 | pL |
| $[Cl]_{in.eq}$ | 6 | mM |
| $[Cl]_{out}$ | 110 | mM |
| $\tau_k$ | 5 | second |
| $\Delta_k$ | 0.2 | nS/Hz |
| $\sigma_E$ | 0.02 | Normalized spatial scale |
| $\sigma_I$ | 0.03 | Normalized spatial scale |
| $\gamma$ | 1/6 | |

## Stages of focal seizure evolution

To further dissect the main seizure dynamics, the 2-dimensional model was reduced into a simplified one-dimensional version (*Figure 2*). In this model, a transient excitatory input triggers the establishment of a localized, self-sustaining tonic firing region near the bottom of the space, marking the onset of the model seizure (*Figure 2A*, green diamond). The seizure territory initially expands bidirectionally until all neurons near the bottom of the space are recruited, at which point it can only expand towards the top. Meanwhile, firing rates at the center of the seizure territory slowly decrease. The 'activity bump' eventually collapses and transitions spontaneously into repetitive neuronal bursting, marking the transition from the ictal-tonic to the ictal-clonic stage (*Figure 2A*, green star). Repetitive neuronal bursting occurs sequentially according to its distance from the ictal wavefront, forming the fast inward traveling waves (*Figure 2B*). In comparison to the slowly expanding ictal wavefront, the fast inward waves travel two orders of magnitude faster (*Figure 2B*, ratio = 170). This ratio approximates that previously reported in human recordings, that is average speed 0.83±0.14 mm/sec for ictal wavefront expansion (*Schevon et al., 2012*) versus post-recruitment ictal discharge traveling speeds of 26±24 cm/sec (mean±s.d.) (*Smith et al., 2016*; *Liou et al., 2017*). As

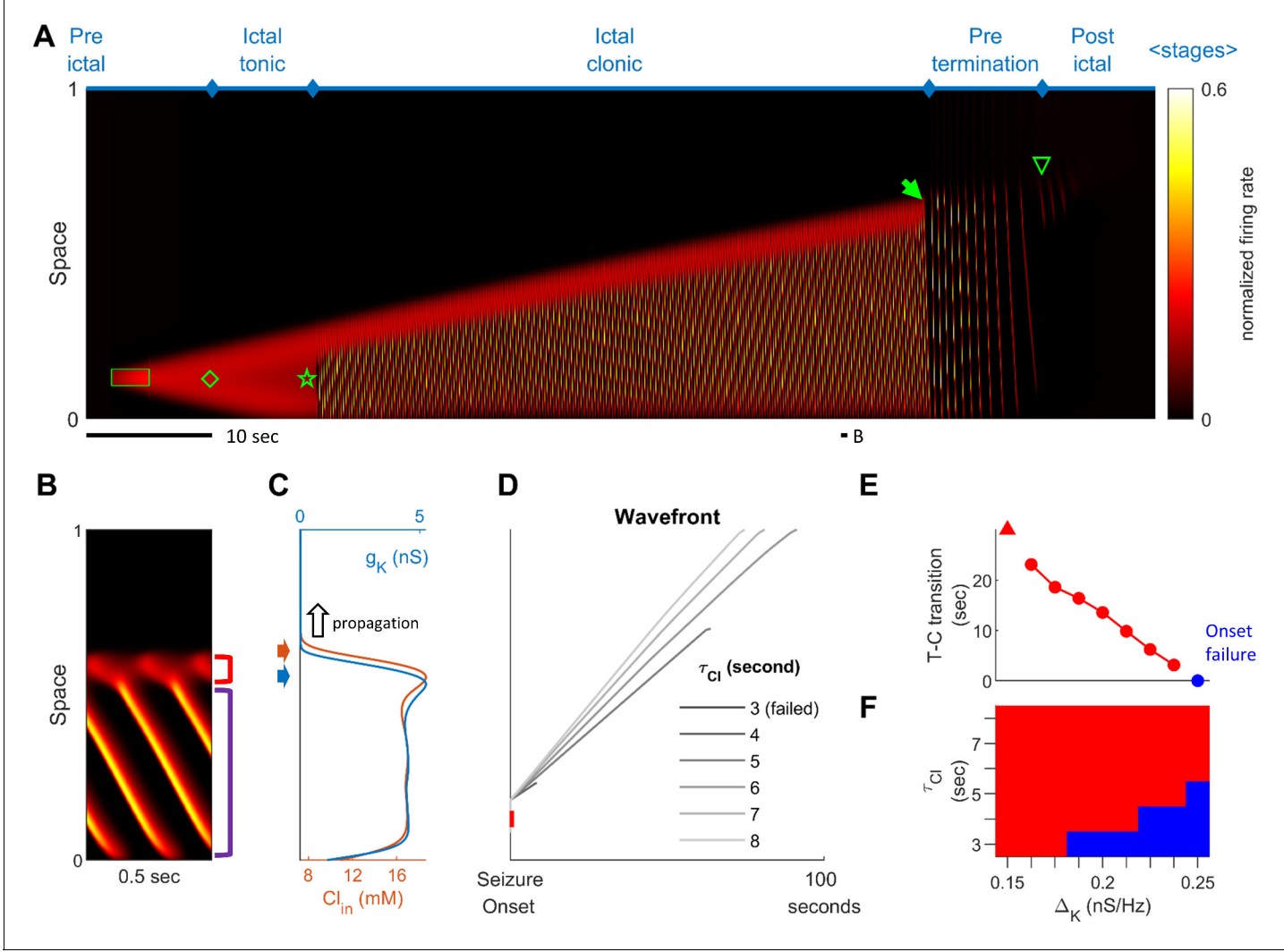

**Figure 2.** Stages of focal seizure evolution in a 1D rate model. (**A**) Spatiotemporal evolution of a model seizure in one-dimensional space ($E_L$=-57.5 mV). Horizontal axis: time; vertical-axis: space. Seizure dynamics is partitioned into the following distinct stages: pre-ictal, ictal-tonic, ictal-clonic, pre-termination, and post-ictal. Green box: seizure-provoking input ($I_d$=200 pA). Green diamond: establishment of external-input-independent tonic-firing area. Green star: tonic-to-clonic transition. Green arrow: annihilation of the ictal wavefront. Green triangle: seizure termination. (**B**) Temporal zoom-in from Panel A during the ictal-clonic stage. Seizure territory further subdivided. Red square bracket: the ictal wavefront. Purple square bracket: the internal bursting domain. Thus, the top/bottom of the space corresponds to outside/inside of the seizure territory. Note that the fast-moving traveling waves move inwardly. Speed: 1.36 normalized space unit per second. Propagation speed of the ictal wavefront: 0.008 normalized space unit per second, for a traveling wave:wavefront speed ratio of 170. (**C**) Intracellular chloride concentration and sAHP conductance as a function of spatial position during the ictal-clonic stage, corresponding to the activity depicted in panel B. Note the distinct spatial fields of the two processes near the ictal wavefront. (**D**) Impaired chloride clearance allows seizure initiation and speeds up seizure propagation. Gray lines: the expansion of border of the seizure territory versus time. End of the lines: seizure termination, either spontaneous ($\tau_{Cl}$≤5 seconds) or after meeting the neural field boundary ($\tau_{Cl}$≥6 seconds). (**E**) Activation of the sAHP conductance leads to the tonic-to-clonic transition. Duration of ictal-tonic stage monotonously decreases as the sAHP conductance increases. Red triangle: persistent tonic stage without transition. (**F**) The seizure-permitting area in $\tau_{Cl}$ and $\Delta_K$ parameter space. Blue/red: onset failure/success.

The online version of this article includes the following figure supplement(s) for figure 2:

**Figure supplement 1.** Stages of focal seizure evolution in a 1D spiking model.

**Figure supplement 2.** Effects of input triggers on model seizure dynamics.

**Figure supplement 3.** Targeted parameter search of seizure territory expansion.

**Figure supplement 4.** Stages of focal seizure evolution in a generalized model of exhaustible inhibition.

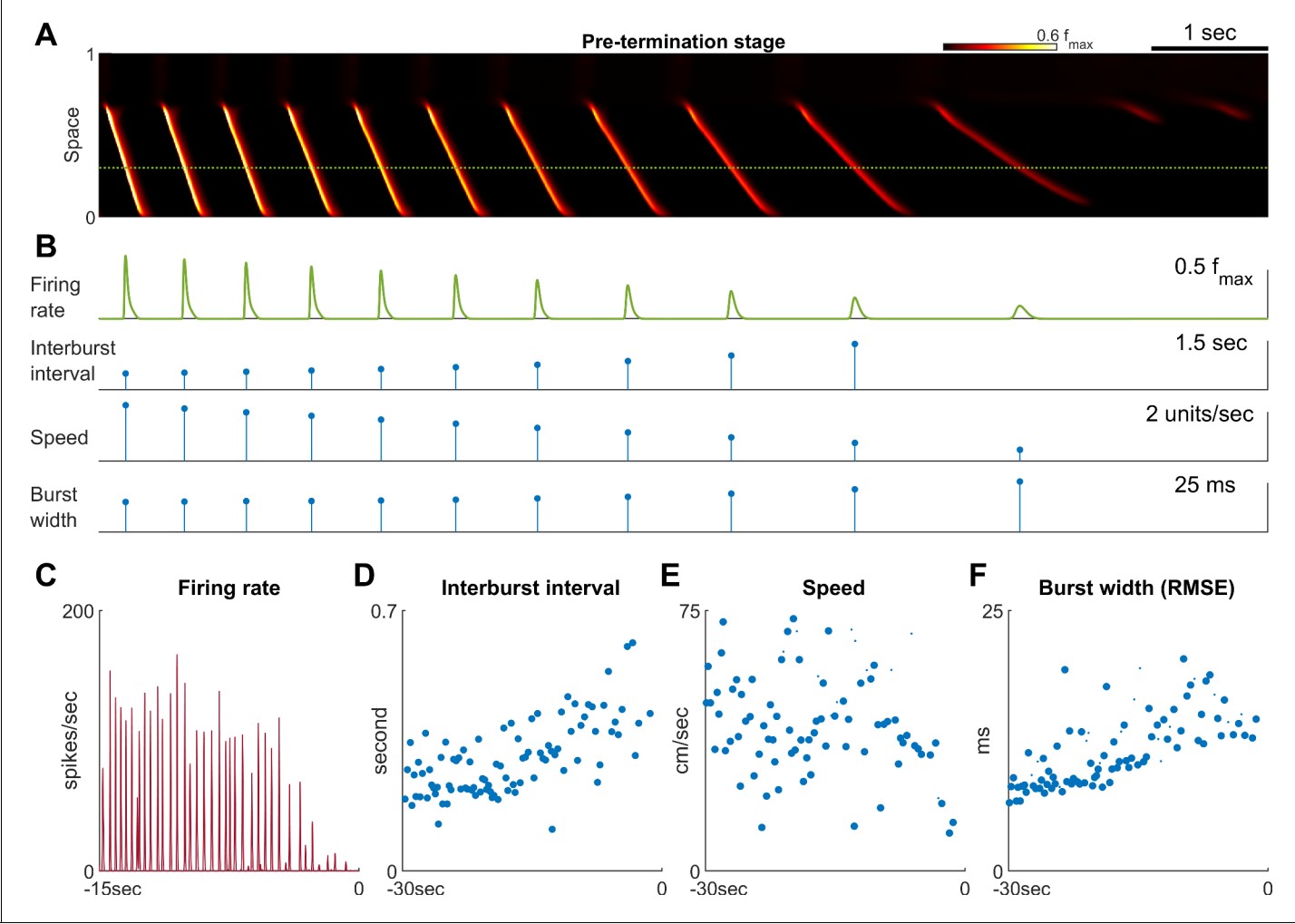

**Figure 3.** Pre-termination stage shows 'slowing-down' dynamics. (**A**) Model seizure, pre-termination stage, zoomed-in from *Figure 2A*. The horizontal axis is shared with Panel B. The local neuronal population dynamics marked by the dotted green line is further shown in Panel B. Notice that in a space-time diagram a traveling wave is a slant band whose slope is its traveling velocity. (**B**) Quantification of the local neuronal population dynamics marked by the dotted green line in Panel A. Peak firing rate decreases, interburst intervals prolong, traveling wave speed decreases, and burst width decreases as the model seizure approaches termination. Neural activity within a region of 0.05 normalized distance centered at the dotted green line is used to calculate traveling wave speed. Burst width is calculated by first treating $f(t)$ during each burst (100 ms temporal window) as a distribution and then estimate its standard deviation. (**C–F**) Patient B shows similar trends of neural dynamics at pre-termination stage. Peak firing rate decreased (**C**) interburst interval increased (**D**), traveling wave speed decreased (**E**), and burst width increased (**F**). Spearman's correlation coefficients: -0.62 (n=35, p<0.001), 0.64 (n=93, p<0.001), -0.34 (n=93, p=0.002), 0.78 (n = 93, p<0.001) for C-F respectively.

the seizure nears termination, the ictal wavefront dissipates (*Figure 2A*, green solid arrow), marking the beginning of the pre-termination stage. Bursts become less frequent and begin to spread out and weaken. After the last burst propagates across modeled brain region, the seizure terminates abruptly (*Figure 2A*, green triangle). The key dynamics of each of these seizure stages are further corroborated and validated in the corresponding 1-dimensional integrate-and-fire spiking model (*Figure 2—figure supplement 1*), that includes threshold noise. The sequence of seizure stages is not affected by duration, spatial extent, or intensity of the external seizure-provoking inputs as long as they are adequate to trigger seizure onsets (*Figure 2—figure supplement 2*). Inputs that are inadequate to initiate a seizure, yet are near enough to the threshold for seizure induction, may trigger short-runs of 'after-discharges' (*Figure 2—figure supplement 2A*).

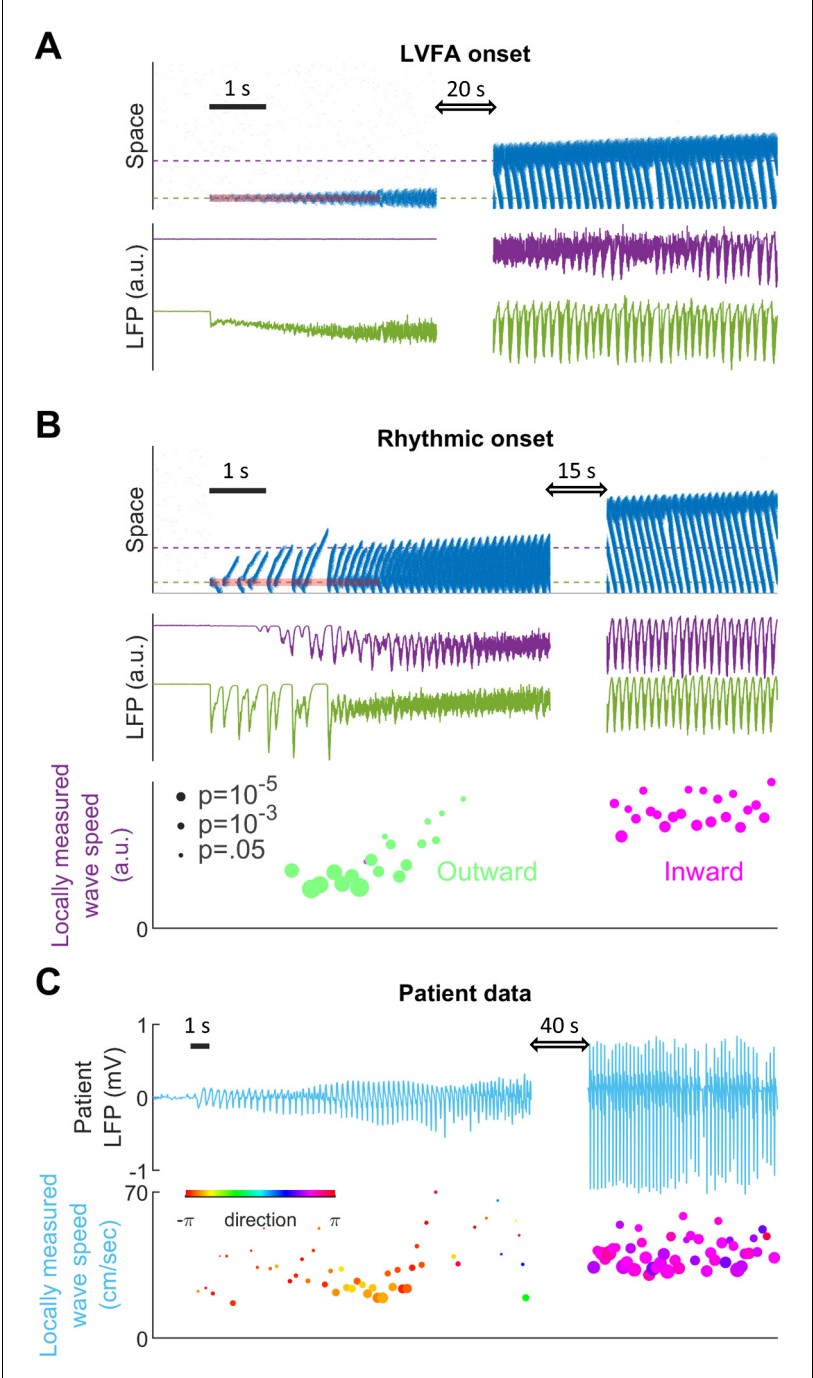

**Figure 4.** Subtypes of seizure onset pattern can arise from different distributions of recurrent inhibition. (**A**) LVFA onset ($E_L$=-60mV, $\gamma$=1/6, $\beta$=1.5 mV). Upper subpanel: raster plot of the spiking model. Horizontal axis: time. Vertical axis: space. Shaded red zone: $I_d$=80 pA. Green and purple dashed lines indicate where the simulated LFPs, shown in the lower subpanel, are read out. LVFA is associated with large DC shift of the LFP, corresponding to seizure onset and ictal wavefront invasion. Periodic LFP discharges emerge after ictal wavefront passage (blank region: 20-second simulation result skipped for visualization). (**B**) Rhythmic onset ($E_L$=-60mV, $\gamma$=1/2, $\beta$=1.5 mV). Upper and middle subpanels inherit the conventions of panel A. At seizure onset, waves are generated and travel outwardly (shaded red area, $I_d$=80pA). They are associated with rhythmic LFP discharges and precede the fast activity. Inward traveling waves emerge after wavefront establishment (after the blank interval). Lower subpanel: comparison of speeds between outward and inward traveling waves. Traveling wave speeds are measured locally at the location indicated by the purple dashed line, with each dot's X-coordinate as the time of local firing rate

*Figure 4 continued on next page*

*Figure 4 continued*

max. Dot size corresponds to F-test p-value (significance level: 0.001) and color represents the direction of propagation. Outward waves (n=34) travel at a significantly lower speed than inward waves (n=23) (U-test, p<0.001). (**C**) Rhythmic seizure onset recorded from Patient B. Upper subpanel: averaged LFP recorded from the microelectrode array. Lower subpanel: traveling wave direction (dot color) and speed (Y-coordinates), estimated according to the spatiotemporal distribution of multiunit spikes. The seizure started with periodic LFP discharges before fast activity emerged. As the seizure evolved, traveling wave direction switched (dot color: orange to purple) and the speed increased (U-test p<0.001, n = 67 versus 49), analogous to the outward-to-inward switch predicted in Panel B.

The online version of this article includes the following figure supplement(s) for figure 4:

**Figure supplement 1.** Local environment determines seizure onset, propagation and termination patterns.

## Physiology of the ictal wavefront

The slow propagation of the modelled ictal wavefront results from two sequentially activated processes: usage-dependent exhaustion of inhibition and upregulation of adaptation currents (*Figure 2C*). These are modeled as intracellular chloride accumulation and sAHP conductance activation, respectively.

In modeled seizures, surround inhibition initially blocks the outward advance of the ictal wavefront. The sequence of events in the transition to seizure is depicted in *Figure 2B–C*. As the ictal wavefront approaches (*Figure 2B*, red bracket and *Figure 2C*, black arrow), intracellular chloride starts to accumulate just ahead of it (brown arrow) due to strong feedforward inhibition projected from the ictal core. Intracellular chloride accumulation causes the chloride reversal potential to become less negative, which compromises the strength of GABA-A receptor-mediated inhibition (Materials and methods, *Equation 3*). As the chloride reversal potential becomes less negative, inhibition is eventually overcome, causing neurons to transition from the resting to the tonic-firing state as they join the ictal wavefront. Intense neuronal firing subsequently activates the sAHP conductance (*Figure 2C*, blue arrow) (Materials and methods, *Equation 4*), which mediates a hyperpolarizing potassium current that curbs neuronal excitability. Consequently, the tonically firing neurons in the ictal wavefront transition into repetitive bursts alternating with periods of silence, forming the periodic inward traveling waves described above. The sequence of intracellular chloride accumulation followed by the activation of sAHP thus mediates both the slow propagation of the ictal wavefront and the periodic, inward fast traveling waves that characterize the ictal core.

The robustness of inhibition and the amount of adaptation conductance both control the degree to which the model is prone to seize. Model cortices in which neurons can pump out chloride quickly are resistant to seizure invasion. As shown in *Figure 2D*, speeding up chloride clearance (low $\tau_{Cl}$) results in a slower ictal wavefront propagation speed, shorter seizure duration, and therefore a smaller seizure territory (also see *Figure 2—figure supplement 3*). Ultimately, if chloride can be pumped out fast enough, the model is unable to form self-sustaining seizures (*Figure 2D*, $\tau_{Cl} \leq 3$ sec). The adaptation conductance ($\Delta_K$) also curbs seizures by preventing the establishment of an ictal core and hastening seizure termination. As shown in *Figure 2E*, increasing the sAHP conductance results in earlier tonic-clonic transitions, earlier seizure termination, and therefore a smaller seizure territory. For high levels of the sAHP conductance, seizure initiation fails (*Figure 2E*, $\Delta_K \geq 0.25$ nS/Hz). *Figure 2F* shows the region in the parameter space of $\tau_{Cl}$ and $\Delta_K$ that permits the formation of an ictal core. Additional series of parameter searches are summarized in *Figure 2—figure supplement 3*. In general, inhibition, which can be strengthened by low intracellular equilibrium chloride concentration ($\left[Cl_{in.eq}\right]$), high chloride buffer capacity ($V_{d,Cl}$), and fast chloride clearance ($\tau_{Cl}$), restrains ictal wavefront propagation. Factors that amplify adaptation currents, including sAHP conductance ($\Delta_K$), low reversal potential of potassium ($E_K$), and fast sAHP activation ($\tau_K$), play a less significant role in modulating the wavefront propagation speed. Instead, they facilitate transitions of seizure stages. In a parameter regime with robust inhibition and strong adaptation, the ictal core cannot be established, making the models resistant to the seizure-provoking insults.

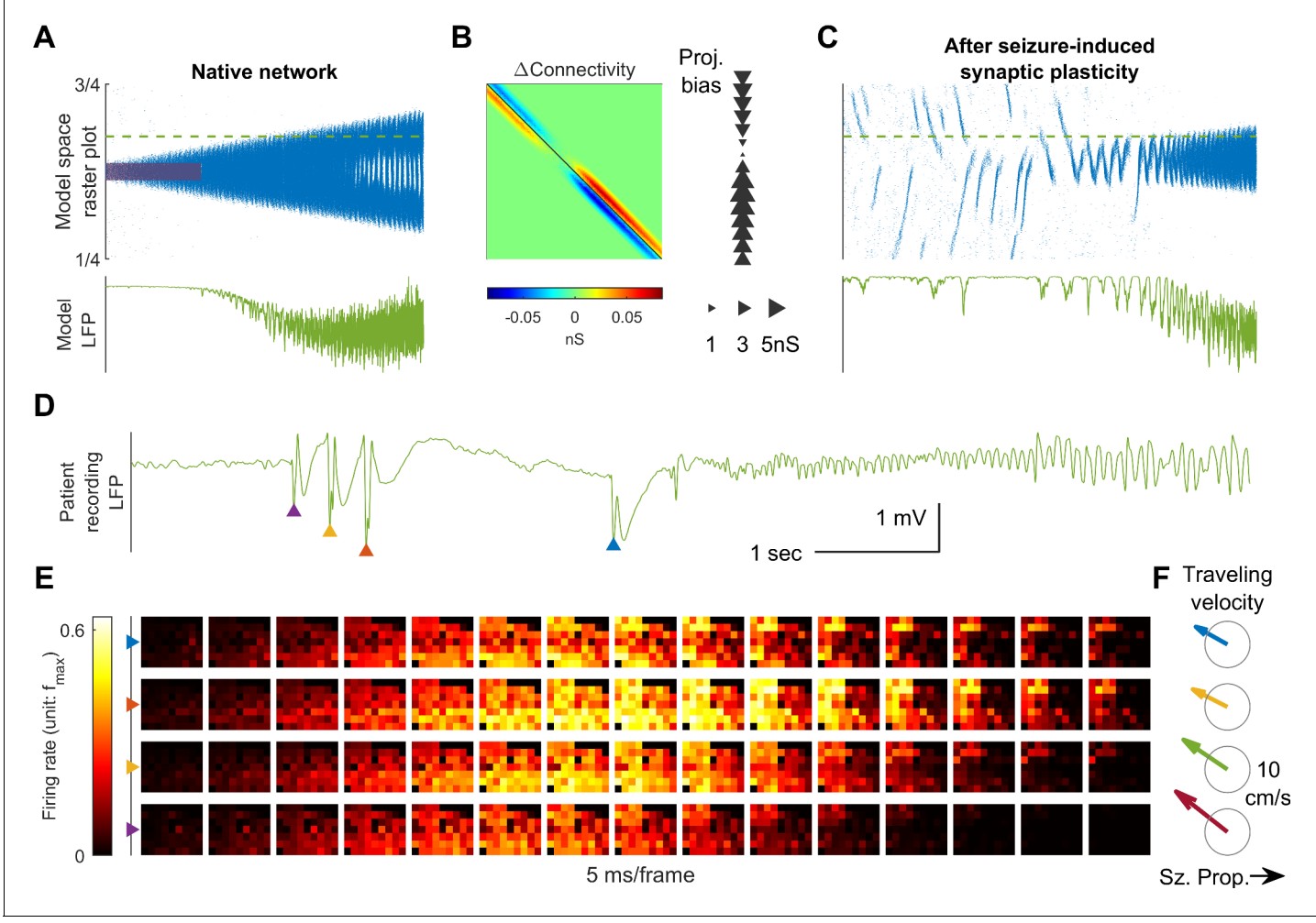

**Figure 5.** Connectivity induced by spike-timing dependent plasticity may promote emergence of seizure foci. (A) Raster plot of a provoked model seizure with LVFA onset pattern. Figure conventions are inherited from *Figure 4A*. Red shaded area indicates the epileptogenic input ($I_d$=200 pA for 3 seconds). Stochastic background input: $\sigma_s$=20 pA, $\lambda_s$=0.1, $\tau_s$=15 ms. The vertical axis is normalized spatial scale and aligned with Panel **B** and **C**. The green dashed line indicates where the simulated LFP, shown in the lower subpanel, is calculated. ($\Delta_K$=0.05 nS, $\beta$=1.5 mV) (B) Changes in recurrent excitatory connectivity strength, $W_E$, induced by STDP after the provoked model seizure (rows correspond to postsynaptic labels, columns to presynaptic). Left subpanel: the $\Delta W_E$ matrix. Right subpanel: neuronal projection bias calculated from the $\Delta W_E$ matrix (see Materials and methods). The location, size, and direction of the triangles represent the position of the neuron, magnitude and direction of the neuron's projection bias respectively. (C) A spontaneous seizure after seizure-induced synaptic plasticity ($I_d$=0 pA). Several large amplitude LFP discharges, shown in the lower subpanel, precede the seizure onset. These LFP discharges are generated by the centripetal traveling waves seen in the raster plot. (D) LFP discharges recorded immediately before Patient A's seizure onset. Discharges are marked by triangles of different colors. Evolution of the associated multiunit firing pattern is shown in Panel E. (E) Multiunit firings constitute traveling waves (10 ms kernel) before LVFA seizure onset. Note that the right half of the array detected multiunit firings earlier than the left half, which is opposite to the expansion direction of the ictal core (left to right). (F) Estimated traveling wave direction and speed. Gray circle: 10 cm/sec.

## Ictal wavefront annihilation and the pre-termination stage

Previously, based on analysis of human multiscale recordings, our group proposed that dissipation of the ictal wavefront is a key event preceding wide-area, simultaneous seizure termination (*Smith et al., 2016*; *Liou et al., 2017*). This process is also evident in the model seizures. As the seizure territory expands, the ictal wavefront in the model encounters increasingly stronger inhibition ahead of itself. Such stronger inhibition is a result of non-localized recurrent inhibition (the global part of *Equation 5* in Materials and methods) becoming progressively activated as more cortical areas are recruited into the seizing territory. The dynamics thus function as a spatial integrator, with strength of inhibition proportional to the extent of the area that has been invaded. Consequentially,

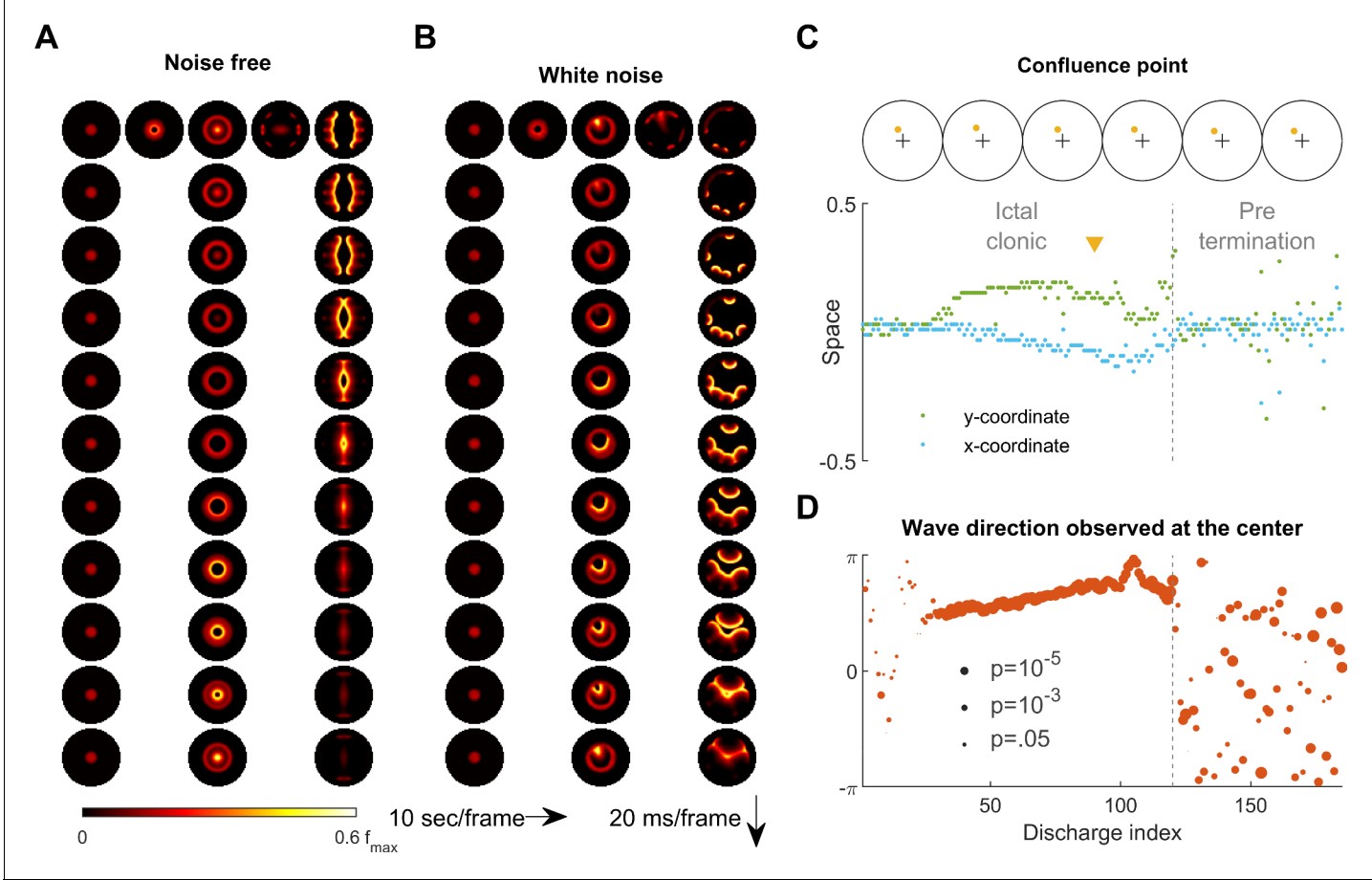

**Figure 6.** Consistent inward traveling wave direction can emerge from white noise. (A) Simulation results of a spatially bounded, two-dimensional rate model under a noise free condition. The epileptogenic stimulus was provided at the center of the round field ($I_d$=200 pA, 3 second, stimulation radius = 0.05). Seizure territory slowly expands (the top row, left to right, 10 seconds per frame) and evolves in stages. The 1st column shows the tonic stage during which the firing rate within the seizure territory stays largely constant (20 ms per frame vertically). The 2nd and 3rd columns show the clonic and pre-termination stages, respectively. Because the neural sheet is symmetric and noise-free, all inward traveling waves meet exactly at the center. (B) Simulation protocol as in panel A but with spatiotemporally white noise with diffusion coefficient 200 pA$^2$/ms is injected uniformly over the whole neural sheet. The confluence point of inward traveling waves during the clonic stage deviates from the center (middle column). Complex, asymmetric traveling wave generation is observed during the pre-termination stage (right column). (C) Locations of confluence points, quantified from Panel B. The upper row: confluence point locations of 6 consecutive inward traveling waves (marked in yellow). The lower subpanel: evolution of the confluence point locations. During the clonic stage (n=120), confluence points significantly deviate from the center (signed-rank test for both $x$ and $y$ coordinates, p <0.001) and drift slowly (auto-correlation coefficients: $\rho_x(1)$=0.82, $\rho_y(1)$=0.83; both p < 0.001.). (D) Traveling wave direction estimated at the center (radius = 0.05). White noise generates a consistent traveling wave direction during the clonic stage once the confluence point drifts away from the center. Circular auto-correlation $\rho_\theta(1)$ during the ictal clonic stage: 0.97, p<0.001, n = 120. However, traveling wave direction becomes more variable during the pre-termination stage: $\rho_\theta(1)$=0.09, p=0.5, n = 65.

The online version of this article includes the following video and figure supplement(s) for figure 6:

**Figure supplement 1.** Increasing traveling wave direction variability as Patient B's seizure approached termination.

**Figure 6—video 1.** Full evolution of the model seizure shown in *Figure 6A*.
https://elifesciences.org/articles/50927#fig6video1

**Figure 6—video 2.** Full evolution of the model seizure shown in *Figure 6B*.
https://elifesciences.org/articles/50927#fig6video2

seizure propagation gradually slows as the wavefront expands. Once the propagation speed is inadequate to escape from sAHP-induced suppression, the wavefront is annihilated, and the seizure transitions into the pre-termination stage (*Figure 2A* green arrow and *Figure 3*).

The pre-termination stage of a model seizure (*Figure 3A*) shows four characteristic trends: decreasing peak firing rate, increasing inter-burst intervals, decreasing inward traveling wave speed,

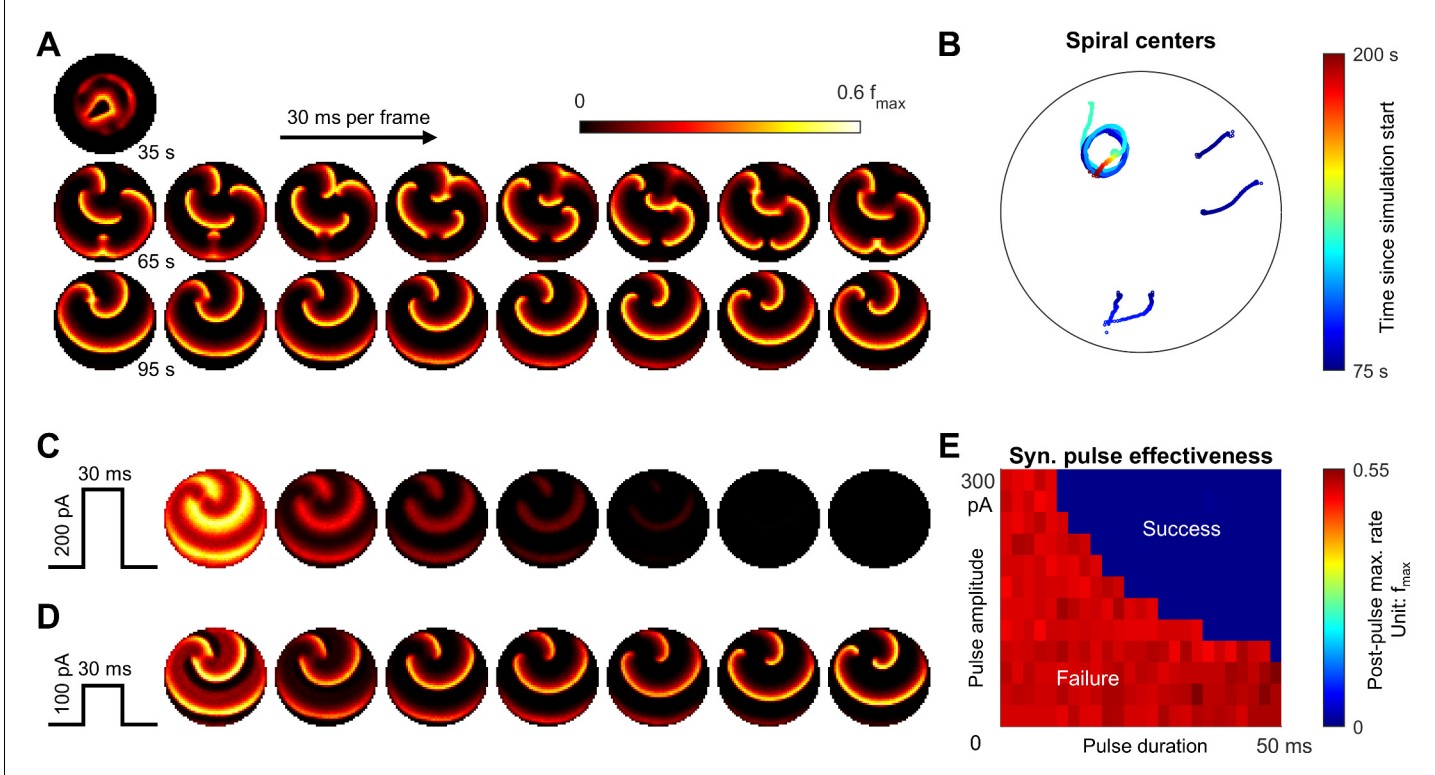

**Figure 7.** Emergence of spiral waves, status epilepticus, and seizure termination induced by globally synchronizing inputs. (A) Spiral wave formation. Model parameters and figure conventions are inherited from *Figure 6B*. The top snapshot shows the ical-clonic stage, then several spiral waves emerged after ictal wavefront annihilation (the upper row of snapshots). Spiral waves demonstrate complex interactions. Some spiral centers survived and persisted indefinitely (the lower row). (B) Movement of spiral wave centers, quantified from Panel A. In this simulation, one spiral wave eventually dominated the whole field and never terminated. (C) A globally synchronizing pulse with adequate amplitude and duration ($I_d$=200pA, 30 ms) forced the spiral-wave seizure to terminate. The pulse was given at the time of the left lower sub-panel in Panel A. Each subpanel is aligned with the lower row of Panel A for comparison. (D) As C, but with inadequate amplitude ($I_d$=100pA) to terminate the seizure. (E) Results of the pulse synchronization study. Color code: maximal firing rate across the neural network within 3 s after the pulse.

The online version of this article includes the following video for figure 7:

**Figure 7—video 1.** Full evolution of the model seizure shown in *Figure 7*.
https://elifesciences.org/articles/50927#fig7video1

and increasing burst width (*Figure 3B*). These 'slowing-down' trends match clinical observations as well as microelectrode recordings from epilepsy patients (*Figure 3C–F*, *Source data 2*; *Source data 3*; *Source data 4*). Increased duration of the silent period following each burst (*Figure 3D*) allows more time for recovery of inhibition strength, as more chloride can be removed from the intracellular space to restore the transmembrane chloride gradient. Recovered inhibition then attenuates burst intensity (*Figure 3C*), slows the speed of the inward traveling waves (*Figure 3E*), and desynchronizes the neuronal population (*Figure 3F*).

## Generalized model for exhaustible inhibition

Although pathological intracellular chloride accumulation is an appealing candidate mechanism for usage-dependent exhaustion of inhibition, other mechanisms have been proposed such as depolarization block of inhibitory neurons (*Ahmed et al., 2014*; *Meijer et al., 2015*), and other pathways that may not be limited to cortical neuronal structures may also exist. We hypothesized that any mechanism that compromises inhibition strength, operates over a time scale of seconds and is triggered by intense inhibition usage is a possible candidate mechanism for the slow expansion of both the seizure territory and the pre-termination dynamics. To confirm our hypothesis, we modeled a generalized inhibition exhaustion process instead of using the chloride accumulation assumption

(Materials and methods, *Equation 8*). All evolutionary stages of focal seizures and their characteristic dynamics were reproduced using this generic process (*Figure 2—figure supplement 4*).

## Distribution of recurrent inhibitory projections determines seizure onset patterns

We have thus far shown that minimal, neurophysiologically based assumptions can account for the complex spatiotemporal evolution of focal seizures. Next, we replaced rate-based units in our one-dimensional model with integrate-and-fire neurons (*Table 2*). LFP-like signals were read out using a computationally validated approximation based on empirical cortical circuit configurations and somatodendritic orientations (*Mazzoni et al., 2015*). Briefly, local synaptic currents are summed, with the contributions from inhibitory synaptic currents scaled up and delayed in time (see Materials and methods). This allowed us to compare the LFP-like signals generated from the model with LFPs from clinical recordings.

Results from the spiking model provide insights into the mechanisms of different focal seizure onset patterns. Low-voltage fast activity (LVFA) is the most common focal seizure onset pattern, with rhythmic discharges or slow rhythmic oscillations seen less frequently (*Perucca et al., 2014*). In our model, seizure onset patterns are determined by the relative strength between local and global recurrent projections (*Figure 4* and *Figure 4—figure supplement 1*). When recurrent projections

**Table 2.** Spiking model parameters.

Parameters are chosen as described in the legend of *Table 1*. Time constant of spike timing dependent plasticity is chosen according to previously reported data (*Bi and Poo, 1998*; *Song et al., 2000*).

| Parameters | | unit |
|---|---|---|
| $C$ | 100 | pF |
| $g_L$ | 4 | nS |
| $g_E$ | 100 | nS |
| $g_I$ | 300 | nS |
| $E_L$ | -57 | mV |
| $E_E$ | 0 | mV |
| $E_K$ | -90 | mV |
| $f_0$ | 2 | Hz |
| $\beta$ | 2.5 | mV |
| $\tau_{ref}$ | 5 | ms |
| $\tau_E$ | 15 | ms |
| $\tau_I$ | 15 | ms |
| $\tau_\phi$ | 100 | mV |
| $\phi_0$ | -55 | mV |
| $\Delta_\phi$ | 2.5 | mV |
| $\tau_{Cl}$ | 5 | Second |
| $V_d$ | 0.24 | pL |
| $[Cl]_{in.eq}$ | 6 | mM |
| $[Cl]_{out}$ | 110 | mM |
| $\tau_k$ | 5 | Second |
| $\Delta_k$ | 0.04 | nS |
| $\sigma_E$ | 0.02 | Normalized spatial scale |
| $\sigma_I$ | 0.03 | Normalized spatial scale |
| $\gamma$ | 1/6 | |
| $\tau_{STDP}$ | 15 | ms |

favor strong localized inhibition, seizure onset is characterized by a localized tonic-firing area, resulting in focal LVFA (*Figure 4A*, 5:1 ratio of local versus global recurrent inhibition, Materials and methods, *Equation 5*). The onset pattern shifts to rhythmic discharges when the distribution of recurrent inhibition favors spatially non-specific projections (*Figure 4B*, 1:1 ratio of local versus global recurrent inhibition, Materials and methods, *Equation 5*). Under this latter condition, inadequate local inhibitory restraint allows ictogenic perturbations to spread quickly outward in the form of traveling waves, generating rhythmic discharges as typically observed in the LFP bands.

Our model further predicts that various LFP patterns may coexist during a single seizure episode, depending on regional variances in the distribution of recurrent connectivity (*Figure 4—figure supplement 1*). Also, our model predicts that outward traveling waves move at a lower speed than inward traveling waves because inhibition outside the ictal wavefront is initially intact (*Figure 4B*). In agreement with our prediction, as shown in *Figure 4C*, a similar dependence of traveling wave speed on the direction relative to the ictal wavefront has been observed in human recordings (*Source data 2*; *Source data 3*; *Source data 4*; *Smith et al., 2016*; *Liou et al., 2017*).

## Seizure-induced network remodeling and spontaneous seizure generation

Seizures have been shown to remodel neural networks (*Scharfman, 2002*; *Elger et al., 2004*; *Lenck-Santini and Scott, 2015*). We therefore incorporated synaptic plasticity into our model to study how seizures affect network connectivity and, subsequently, seizure dynamics. We subjected the excitatory synaptic projections in our 1D spiking model to spike-timing dependent plasticity (STDP, Materials and methods, *Equation 6-7*). During a provoked seizure (*Figure 5A*), inward traveling waves, because of pre-before-post potentiation in STDP, selectively enhanced excitatory projections from the periphery to the internal domain of the ictal core, in accordance with the predominant traveling wave direction (*Figure 5B*, left). This newly created spatial bias of excitatory projections therefore follows a centripetal pattern (*Figure 5B*, right, see Materials and methods, section Spike-Timing Dependent Plasticity for quantification of spatial biases of synaptic projections). Such strengthening of centripetal connectivity can predispose the neural network to more seizures by funneling background neural activity into its center. Furthermore, as shown in *Figure 5C*, a background input which is non-ictogenic in a naïve network may now provoke seizures in the remodeled network by evoking centripetal traveling waves, which can exhaust inhibition strength at the center of the increased centripetal connectivity. Subsequent seizures are therefore prone to be triggered from the same location, and their associated inward traveling waves can further reciprocally strengthen this centripetal connectivity.

Results from seizures generated by our model after the network remodeling described above predict that epileptiform discharges preceding LVFA seizure onset travel toward the seizure center, rather than away from it (*Figure 5C*). Such centripetal traveling waves have been observed in the microelectrode recordings of spontaneous human seizures (*Source data 1*) immediately before the onset of an LVFA seizure (*Figure 5D–F*, compared to *Figure 1B*). In particular, large-amplitude epileptiform discharges with associated multiunit bursts were present just prior to seizure onset, spreading from right to left (*Figure 5E–F*). This is the same direction as the rapidly traveling inward waves of ictal discharges, and opposite to the direction of seizure expansion (*Figure 1B*).

## Variability of traveling wave direction under noisy conditions

We next return to the primary 2D rate model to examine the effects of background noise on the spatiotemporal dynamics of seizures. Under the noise-free condition considered up to this point, a seizure evoked at the center of the neural sheet expands with a perfectly circular ictal wavefront (*Figure 6A* and *Figure 6—video 1*). All inward traveling waves in this case are directed in a centripetal pattern uniformly towards the center point, with no directional preference within the ictal core (*Figure 6A* and *Figure 6—video 1*). However, ictal EEG discharges recorded from epilepsy patients clearly demonstrate stable preferred traveling wave directions (*Smith et al., 2016*; *Liou et al., 2017*; *Martinet et al., 2017*). Although preferred directions can arise from pre-existing spatial asymmetry (*Figure 1—video 1*), our simulation of a seizure occurring at the round, symmetric environment indicates that the traveling wave direction bias may also occur due to random fluctuations in background neural activity (*Figure 6—video 2*).

It might seem that adding random perturbations to our model would make inward traveling waves flip randomly, which would further contradict in vivo observations of a consistent preferred direction in the ictal-clonic stage (*Figure 6—figure supplement 1*; *Smith et al., 2016*; *Liou et al., 2017*; *Martinet et al., 2017*). Instead, our model shows that uniformly distributed, spatiotemporally white noise can create a long-lasting, preferred inward traveling wave direction without any requirement for spatial asymmetry or inhomogeneity of the neural sheet (*Figure 6B*, also see *Figure 6—video 2*). After the tonic-to-clonic transition in the model, background noise first randomly creates a bias of the confluence point where the inward traveling waves meet (*Figure 6C*). Once established, the bias persists and, due to the low pass property of the neural sheet, it can only slowly drift, resulting in a sustained traveling wave direction preference lasting until the pre-termination stage. After wavefront annihilation, traveling wave directions randomly fluctuate and flip (*Figure 6D*). Such variable wave directions mimic human seizure recordings, where an increase in the variability of wave directions and 'flip-flop' direction reversals are present before seizure termination (*Figure 6—figure supplement 1*; *Trevelyan et al., 2007a*).

## Spiral wave formation, status epilepticus, and synchronization-induced termination

The complex topology of inward traveling waves after ictal wavefront annihilation suggests a candidate scenario that can lead to persistent seizure activity (*Figure 7A*). We repeated the simulation shown in *Figure 6B* under the same level of background current noise. Three out of ten repeated simulations showed failure of spontaneous seizure termination. The persistent seizure scenario is characterized by the formation of spiral waves that emerge after wavefront annihilation (*Figure 7A*) and exhibit complex interactions (*Figure 7B*). In the example shown, one spiral wave eventually dominated the space, self-attracted, and persisted indefinitely, resulting in model status epilepticus (*Figure 7B* and *Figure 7—video 1*). The coexistence of endpoints corresponding to spontaneous seizure termination or status epilepticus under background current noise suggests bistable dynamics of seizure evolution (*Kramer et al., 2012*).

Sustained spiral-wave activity, that is status epilepticus in our model can be terminated by a globally projecting excitatory input that is adequately strong to briefly activate the whole neural sheet at once (*Figure 7C–E* and *Figure 7—video 1*). Within hundreds of milliseconds, the spiral waves spontaneously terminate without going through pre-termination slowing of the discharge pace.

## Discussion

We have presented a biophysically constrained computational model of seizures that, despite being limited to a reduced set of neural properties, reproduces many key aspects of human focal seizures. We showed that the collapse of inhibition strength ahead of the ictal wavefront, followed by the emergence of hyperpolarizing currents, accounts for the slow expansion of the ictal core, the tonic-to-clonic transition, and the experimentally observed generation of inward traveling waves. These features are closely aligned with properties of human seizures that were previously described using combined clinical EEG and microelectrode recordings (*Smith et al., 2016*). The interplay of usage-dependent inhibition exhaustion and adaptation, which drives wavefront propagation during early stages of the seizure life cycle, also characterizes seizure-prone versus seizure-resistant tissues. Our study supports the idea that recurrent inhibition that is spatially non-specific terminates seizures, creating the slowing frequency of discharging and increased traveling wave variability during the pre-termination stage. Additionally, the model demonstrates that distinct seizure onset patterns can result from topologically variant recurrent inhibitory projections. The inclusion of STDP in the model results in progressive, localized pathological enhancement of excitatory connectivity, reducing seizure threshold in the model and enabling spontaneous seizure generation. Our results on plasticity emphasize the importance of the discovery and modeling of rapid inwardly directed traveling waves (*Smith et al., 2016*; *Liou et al., 2017*), as these may induce plasticity that increases susceptibility to future seizures. Finally, the model provides candidate scenarios for both spontaneous seizure termination and ongoing status epilepticus, based on stochastic events following dissipation of the ictal wavefront.

The spatial seizure topology inferred from human multiscale recordings (*Schevon et al., 2012*; *Smith et al., 2016*) and explicitly reproduced in the computational model described here has not

previously been studied in its entirety. The propagation of the ictal wavefront is often invisible to standard EEG, and even to wideband microelectrode recordings (*Schevon et al., 2012*). In contrast, the fast-moving inward traveling waves manifest as field potential discharges, forming the classic EEG signature of seizures (*Smith et al., 2016*; *Martinet et al., 2017*). Due to the speed at which these discharges travel (up to 100 cm/sec) (*Liou et al., 2017*), they appear to be synchronized across large brain areas, leading to suppositions of large-area network origin (*Kramer et al., 2010*; *Jirsa et al., 2014*; *Khambhati et al., 2016*). Our model demonstrates that these spatiotemporal dynamics can result from well-established neuronal processes occurring at the level of localized cellular interactions. Further, persistent traveling wave direction preferences can result from random fluctuations in background neural activity. A caution applies, however, in interpreting effects that occur near the boundary of the neural sheet, as this area may be artificially hyperexcitable in the model due to reduction in the maximal inhibitory conductances.

Without modification by STDP, the network is reminiscent of a healthy, non-epileptic brain – it maintains a non-trivial baseline firing rate, does not spontaneously seize, but can be provoked into a full-blown seizure. Varying the external stimulus' strength produces a range of responses from post-stimulation depression, short-run after-discharges, to full-blown seizures. In this sense, perhaps the most straightforward real-world interpretation of the external stimuli in our simulations is the focal electrical zap given during intra-operative cortical mapping. The suprathreshold stimuli are therefore analogous to the electrical shock given during convulsive therapies (*Spellman et al., 2009*). More generally, the stimuli are qualitatively equivalent to any factor that cause a neighborhood of neurons to depolarize, such as anoxic depolarization caused by local ischemia (*Somjen, 2001*) or discharges triggered by transcranial magnetic stimulation (*Lisanby et al., 2003*). Accordingly, responses to the external stimuli, in return, may reveal the network's intrinsic propensity to seize.

Both modern in vitro brain slice studies (*Trevelyan et al., 2007b*) and in vivo human recordings (*Schevon et al., 2012*) support the classical hypothesis of surround inhibition (*Prince and Wilder, 1967*), which motivated the Mexican-hat pattern of connectivity and the usage-dependent exhaustible inhibition mechanism employed in our study (*Eissa et al., 2017*; *Liou et al., 2018*). Theoretically, massive GABAergic activity provoked by ictal events can overwhelm chloride buffering and clearance mechanisms. The collapsed transmembrane chloride gradient compromises the strength of surround inhibition, leading to seizure onset and propagation (*Lillis et al., 2012*; *Alfonsa et al., 2015*). Indeed, excitatory effects of GABAergic transmission have been found in brain slices taken from epilepsy patients (*Cohen et al., 2002*). The paradoxical excitatory effects of GABA in epilepsy patients may be attributed to clearance defects, as tissues from temporal lobe epilepsy (*Huberfeld et al., 2007*) and tumor-associated epilepsy patients (*Pallud et al., 2014*) both showed reduced KCC2 for chloride extrusion. Experimentally knocking down KCC2 also leads to epileptiform discharges (*Zhu et al., 2005*), and an in-vitro fluorescence study showed that chloride accumulation preceded seizure onset (*Lillis et al., 2012*). In agreement with previous studies, our model suggests that tissues with slow chloride clearance are seizure-prone. Inhibition robustness is critical for preventing, restraining, and shaping focal seizures.

The tonic-clonic transition has previously been theorized to arise from intrinsic neuronal processes such as sAHP (*Beverlin et al., 2012*). In our model, we further explore the hyperpolarizing current's role throughout a seizure's life cycle, including controlling seizure onset, mediating tonic-clonic transition, and participating in the evolution of pre-termination seizure activities. Clinical studies have confirmed the importance of sAHP in prohibiting epileptic activity. Mutations of KCNQ2 and KCNQ3 channels, which contribute to sAHP currents, are associated with familial neonatal epilepsy (*Tzingounis and Nicoll, 2008*).

We adopt a simplistic approach to ionic dynamics during seizures. Ion concentrations are held constant except intracellular chloride, which is modelled as the mechanism underpinning usage-dependent exhaustion of inhibition. This model's approach, however, should not be misinterpreted. Other than chloride, significant ionic shifts, including sodium, potassium, calcium, hydrogen and etc., have also been observed during peri-ictal periods (*Raimondo et al., 2015*). The tortuosity of extracellular space further complicates the picture, as ions may distribute unevenly due to diffusion limits (*Syková and Nicholson, 2008*). Dysregulation of neuronal excitability due to extracellular potassium accumulation, for example, has been proposed a candidate mechanism of seizure initiation (*González et al., 2019*). A potential extension of this model may include multiple ion species and their regional variance.

In our model, spontaneous seizure termination is caused by progressive build-up of inhibition, which is used to model the widespread effect of focal seizures (*Burman and Parrish, 2018*; *Liou et al., 2019*). This widespread inhibition has not been extensively described, but nevertheless has been demonstrated in both an acute animal seizure model (*Liou et al., 2018*) and in a computational model validated with human microelectrode recordings (*Eissa et al., 2017*). Experimentally, calcium imaging has confirmed seizure-induced cross-areal activation of PV(+) interneurons, for example in visual cortex in response to a focal seizure triggered in somatosensory cortex (*Wenzel et al., 2017*; *Liou et al., 2018*). Anatomically, such widespread inhibition may be mediated by long-distance, cross-areal projections that preferentially project to inhibitory interneurons in their target circuits (*Zhang et al., 2014*; *Sun et al., 2019*). In addition, focal seizures may depress brainwide activities via disrupting subcortical structures (*Feng et al., 2017*) and ascending activation systems (*Blumenfeld, 2012*). Large-scale non-neural mechanisms that are not specifically modelled in our study may also contribute to seizure termination. For example, global hypoxia secondary to seizure-induced cardiopulmonary compromise may activate ATP-sensitive potassium channels in pyramidal neurons in a spatially non-specific way (*Ching et al., 2012*), thereby terminating seizures by building resistance ahead of the ictal wavefronts and generating termination dynamics, analogous to the global recurrent inhibition effect modelled in our study. In other words, the global inhibition assumption may be interpreted as a hypothesis that, as a seizure territory enlarges, principle neurons tend to hyperpolarize and are therefore progressively harder to recruit, with the specific neurological mechanisms varying between patients.

In this study, we simplified inhibitory interneurons to reduce model parameter space. This should not be misinterpreted as implying that we mean to diminish the role of interneuron dynamics in seizure evolution. Recent studies have shown that interneurons may participate in seizure initiation (*Librizzi et al., 2017*), and their depolarization block can be critical for seizure propagation (*Eissa et al., 2017*). However, complex inhibitory interneuron dynamics, as shown in our study, may not be required for the replication of key features of seizure dynamics.

The biophysical processes proposed here are certainly not the only mechanisms causing seizure evolution. Instead, they should be considered as representative candidates, with the suggestion that other mechanisms should share the features of the ones we have proposed. Processes that produce similar effects on network dynamics and that operate over the same time scales may also contribute to the generation of seizure dynamics, as exemplified by our phenomenological model, in which the general variable z controls local effectiveness of inhibition rather than the transmembrane chloride gradient. For example, in addition to intracellular chloride accumulation, depolarization block of GABAergic neurons may serve a similar role in contributing to the breakdown of inhibition (*Meijer et al., 2015*). Similarly, aside from hyperpolarizing currents, intense neuronal bursts can inactivate sodium channels, quickly reducing neuronal excitability in a couple of seconds (*Fleidervish et al., 1996*). Additional depressing mechanisms, including excitatory synaptic depletion (*Beverlin et al., 2012*) and inhibitory cell recovery (*Ziburkus et al., 2006*) may collectively contribute to tonic-clonic transitions, slowing, and termination. However, key similarities in seizure dynamics observed across different brain regions and pathological conditions suggest that some mechanisms may be universally present, and these are primary candidates for physiological processes underlying seizure evolution. Developing therapies targeting at these mechanisms could therefore achieve broad spectrum anti-seizure effects.

Our model demonstrates that the spatial distribution of recurrent inhibition can shape neural network dynamics, explaining the variance of focal seizure onset subtypes. Seizures manifesting with focal low voltage fast activity, a positive predictor of seizure freedom following resection of the focus (*Alarcon et al., 1995*), occurred in the setting of relatively strong local inhibition in our model. In contrast, relatively weak local inhibition allows seizure-related activity to spread quickly in the form of periodic outward traveling waves moving ahead of the ictal wavefront (*Figure 4B*). These outward traveling waves result in the EEG appearance of rhythmic discharging at seizure onset. Previous studies have proposed that this 'hypersynchronous' seizure onset pattern depends on long-range cortical connections (*Perucca et al., 2014*; *Weiss et al., 2016*), or increased surrounding tissue excitability (*Wang et al., 2017*). Our model instead highlights the role of localized inhibition and demonstrates that both types of electrographic onset signatures can arise from focal sources, depending primarily on how the ictal foci are established.

The model predicts that physiological plasticity mechanisms (STDP in this model) can be hijacked by pathological seizure dynamics (*Mehta et al., 1993*). The fast inward traveling waves during each seizure create self-enhancing pathological centripetal connectivity, resulting in the emergence of pre-ictal large amplitude LFP discharges that manifest as centripetally propagating waves. Such single or repetitive sharp waves are often seen just prior to LVFA seizure onsets in clinical recordings (*Lee et al., 2000*) and were also present in one of our human microelectrode recordings (*Figure 5D–E*). We also reported similar spatial attraction of bursts in a rodent model during interictal periods (*Liou et al., 2019*). Our model therefore offers a theoretical explanation that synaptic plasticity results in spatial rewiring which increases seizure susceptibility and manifests as interictal discharges or herald spikes that travel towards the site of eventual seizures. This model prediction has potential implications in clinical seizure management. If patterns of interictal discharge traveling waves could be used to reduce uncertainty about the location of subsequent seizures, they could be used as a valuable seizure prediction and diagnostic tool. Furthermore, preventing or reversing traveling wave-induced spatial rewiring may break the pathological, self-enhancing loop, eventually leading to slowdown or reversal of seizure progression.

It has been proposed that seizure evolution has bistable endpoints, and that failure of seizure termination can be a stochastic event (*Kramer et al., 2012*). In the presence of noise, the cortex may become trapped in the spiral wave scenario leading to an indefinitely persistent seizure, that is status epilepticus. Animal experiments, both in brain slice (*Huang et al., 2004*) and in vivo (*Petsche et al., 1974*; *Huang et al., 2010*) have shown that spiral waves can develop in acute seizure models. Multielectrode array recordings have also revealed spiral epileptiform discharges during feline picrotoxin-induced seizures (*Viventi et al., 2011*). Such spiral waves have yet to be detected in humans, where they may serve as a marker of risk for status epilepticus.

Our model predicts such spiral waves may be terminated by a brief synchronizing input. In agreement with this prediction, synchronization has been shown to promote seizure termination in status epilepticus patients (*Schindler et al., 2007*). Analogously, spiral waves also develop during ventricular fibrillation, a fatal form of cardiac arrhythmia. Delivering a brief pulse that widely activates cardiomyocytes at once has been the standard therapy to rescue such patients. This suggests that a stimulation strategy involving wide, synchronous excitatory input delivery may be an effective approach for seizures characterized by spiral waves.

Although microelectrode recording plays an important role in validating our model, the key dynamics which are commonly seen behaviorally and observed in macroelectrode recordings, in our opinion, provides an equal, if not more important, support the generality of this model. The question therefore arises why relatively few human microelectrode recordings have demonstrated the existence of ictal wavefronts. Our model, indeed, provides a straightforward explanation – a microelectrode array not only needs to be positioned at a region that is recruited into the seizure territory but also needs to be close enough to the onset spot. Otherwise, the seizure could have evolved into its pre-termination stage, during which the seizure activity is still slowly propagating but the wavefront has been annihilated.

Finally, this model is not built to exactly duplicate the microelectrode recordings. Hyper-tuning the model parameters, in our opinion, is not fruitful as seizure dynamics vary significantly even within the same patient from one episode to another. Instead, we focus on how the cellular neurophysiological principles generate the key spatiotemporal dynamics in a larger scale. Similar bottom-up approaches have also been adopted to study the effects of network connectivity on seizure initiation and subtypes (*Wang et al., 2017*; *Jacob et al., 2019*). Alternatively, seizure EEG and ECoG databases have allowed a top-down, machine learning-based approach (*Karoly et al., 2018*). The availability of macroelectrode data might reduce patient selection biases. However, making mechanistic inference based on LFP-derived signals is challenging because of their limited resolution and sensitivity to geometric configurations (*Einevoll et al., 2013*). In our perspective, the two mutually complementary approaches are both indispensable in order to fully understand seizure dynamics.

## Conclusions

We found that a surprisingly small number of well-understood, biophysically informed neuronal processes can explain the complex, large-scale spatiotemporal evolution of focal seizures. Our reductionist approach provides insights into generalizable principles underlying complex seizure dynamics, without the need to replicate the seizing brain neuron by neuron. The model's seizures

are notably consistent with both clinical semiology and microelectrode array recordings of human seizure events, including the morphology of the ictal wavefront, distinct stages of pre-recruitment, post-recruitment, and pre-termination, wide-ranging ictal and interictal discharges, development of a fixed, chronic seizure focus, spontaneous seizure termination, and status epilepticus. These parallels provide evidence that the topological pattern of a slowly propagating ictal wavefront with rhythmic inward fast traveling waves should be considered the fundamental topology of neocortical focal seizures. Additional investigation is needed to describe the potential impacts of subcortical involvement, cross-hemisphere interaction, and cross-regional propagation.

# Materials and methods

## Data collection and processing

Electrophysiology data were obtained from patients with pharmacoresistant focal epilepsy undergoing invasive EEG monitoring at Columbia University Medical Center as part of their clinical care, and who were additionally enrolled in a study of microelectrode recordings of seizures (*Schevon et al., 2008*; *Waziri et al., 2009*; *Schevon et al., 2012*). The Columbia University Medical Center Institutional Review Board approved the research (protocol number: AAAB-6324), and informed consent was obtained from participants prior to surgery. Only data from patients whose microelectrode array-sampled cortical area were recruited into the ictal cores were included in the current study (two patients, four seizures) (*Schevon et al., 2012*; *Smith et al., 2016*). Data from patients who developed generalized seizures or whose array-sampled cortical area lack intense, phase-locked multiunit bursts or propagating ictal wavefront (ictal penumbra) was excluded from this study (five patients). Data processing algorithms have been previously published. Briefly, multiunit spikes were extracted from Utah array recordings by filtering the raw data between 300 and 3000 Hz and detecting threshold crossings ($-4$ s.d.) (*Quiroga et al., 2004*). Multiunit firing rate was calculated by convolving multiunit spike trains with Gaussian kernels (10 ms for fast dynamics and 100 ms for examining expansion of seizure territory). All calculations were performed using in-house software (Matlab, Mathworks, Natick, MA).

## Rate model

A spatially homogeneous one-dimensional neural field is evenly discretized into 500 populations. Within each population, a principle neuron, described by the following conductance model, is used to approximate population dynamics:

$$C\frac{\partial V}{\partial t} = \sum_i g_i(E_i - V) + I \qquad (1)$$

where $V$ is membrane potential and $C$ is cell capacitance. Four types of conductances ($g$) are modelled: leak ($g_L$), glutamatergic synaptic ($g_E$), GABAergic synaptic ($g_I$), and slow afterhyperpolarization, sAHP, ($g_K$) conductances. $E_i$ represents each conductance's reversal potential. Neurons receive external current inputs, $I$, coming from outside the neural network. The inputs are divided into deterministic, $I_d$, and stochastic parts, $I_s$. The stochastic part may be spatiotemporally white or colored, with the latter generated by Ornstein-Uhlenbeck processes with amplitude $\sigma_s$ and time and spatial filter constants, $\tau_s$ and $\lambda_s$, respectively. Each population's mean firing rate, $f$, is calculated by passing the difference between the principle neuron's membrane potential, $V$, and its firing threshold, $\phi$, through a sigmoid function, $f(v) = \frac{f_{max}}{1+\exp(-v/\beta)}$, where $v = V - \phi$. $\beta$ controls slope of the sigmoid function, and $f_{max}$ is the population's maximal firing rate. The firing threshold is dynamic with its steady state value linearly dependent on $f$ with coefficient $\Delta_\phi$,

$$\tau_\phi \frac{\partial \phi}{\partial t} = (\phi_0 - \phi) + \Delta_\phi f \qquad (2)$$

where $\phi_0$ represents the principle neuron's baseline threshold.

The reversal potential of GABAergic conductance ($g_I$) depends on the principle neuron's transmembrane chloride gradient. Specifically, $E_{Cl}$ is calculated according to Nernst equation, $E_{Cl} = -26.7 \log \frac{[Cl_{out}]}{[Cl_{in}]}$ mV. Intracellular chloride concentration is determined by two counteracting

mechanisms: chloride current influx and a clearance mechanism with first-order kinetics (*Zhu et al., 2005*),

$$\frac{\partial [Cl]_{in}}{\partial t} = -\frac{I_{Cl}}{V_d F} + \frac{[Cl]_{in,eq} - [Cl]_{in}}{\tau_{Cl}} \tag{3}$$

where $I_{Cl}$ is chloride flow through GABA-A receptors, $I_{Cl} = g_{Cl}(V - E_{Cl})$, $V_d$ is volume of distribution of intracellular chloride (*Marchetti, 2005*; *Vladimirski et al., 2008*), $F$ is Faraday's constant, $\tau_{Cl}$ is the time constant of the chloride clearance mechanism (*Deisz et al., 2011*), and $[Cl]_{in,eq}$ is the equilibrium intracellular chloride concentration. The steady state of the sAHP conductance, $g_K$, is also linearly dependent on firing rate via $\Delta_K$ and evolves according to first-order kinetics,

$$\tau_K \frac{\partial g_K}{\partial t} = -g_K + \Delta_K f \tag{4}$$

with $\tau_K$ is significantly larger than the time constant for threshold adaptation (of order seconds; *Table 1*).

Inhibitory interneurons are simplified in this study. Their membrane potential dynamics is not specifically modeled, and they react instantly and with a monotonic dependence on their synaptic inputs and only project to their corresponding excitatory neurons at the same location. With the help of these instantly reacting interneurons, model neurons are computationally equivalent to emitting both excitatory and inhibitory synaptic projections to each other. Notice that there is no long-range projecting interneurons required (*Figure 1A*; *Bressloff, 2014*; *Dayan and Abbott, 2001*). Recurrent excitation, $g_E$, is computed by first convolving the normalized firing rate, $A = \frac{f}{f_{max}}$, with a spatial kernel, $\int_{-\infty}^{+\infty} A(x - s, t) K_E(s) ds$, where $K_E$ is a zero-mean Gaussian of variance $\sigma_E^2$. The results are also temporally filtered to model synaptic delay (single exponential with time constant $\tau_E$) and then multiplied by the strength of recurrent excitation, $\bar{g_E}$. Recurrent inhibition is calculated analogously and can be partitioned into spatially localized and non-localized parts

$$K_I \sim (1 - \gamma) G(0, \sigma_I^2) + \gamma U \tag{5}$$

where $G$ is a Gaussian, $U$ is spatially uniform over the whole neural field and $\gamma$ controls the relative contribution of each component. Recurrent inhibition extends wider than recurrent excitation ($\sigma_I > \sigma_E$). All models are simulated with dt=1 ms. All the model simulation codes are available at https://github.com/jyunyouliou/LAS-Model (*Liou, 2019*; copy archived at https://github.com/elifesciences-publications/LAS-Model).

## Spiking model

2000 neurons with membrane potentials modeled as in *Equation 1* are evenly distributed along a bounded one-dimensional space. Spikes are emitted stochastically with instantaneous firing rate, $f = f_0 \exp\left(\frac{V - \phi}{\beta}\right)$, where $f_0$ is a parameter and $\beta$ quantifies the 'uncertainty' of action potential threshold. Numerically, spikes are generated by a Bernoulli process within each time step. If a spike is emitted, the average membrane potential at the spike-emitting time step is taken to be the average of the pre-spiking membrane potential and the peak of the action potential (+40 mV). After a spike, the membrane potential is reset to 20 mV below the pre-spike membrane potential. During the following refractory period ($\tau_{ref}$), membrane potentials are allowed to evolve but spiking is prohibited by setting $f = 0$. In analogy with *Equation 2*, the threshold increases by $\Delta_\phi$ immediately after a spike, and it exponentially decays with time constant $\tau_\phi$. Chloride dynamics, sAHP conductance, and recurrent projections are modeled as described in the previous section.

## Spike-timing dependent plasticity (STDP)

For simulations of one-dimensional spiking models in which network remodeling is considered, recurrent excitation is subject to spike-timing dependent plasticity (STDP) (*Song et al., 2000*; *Bi and Poo, 1998*). The recurrent synaptic weight matrix, $W_E$, is partitioned into two parts:

$$W_E = W_C \odot W_P \tag{6}$$

where $W_C$ is the convolution matrix of the Gaussian excitatory kernel ($K_E$), $W_P$ is a matrix

representing synaptic weight adjustment according to the STDP rule, and $\odot$ represents element-wise multiplication (*Toyoizumi et al., 2014*). Every entry of $W_P$ is initiated at 1. After each model seizure, the plastic part of the synaptic projection from neuron $b$ to $a$, namely, the entry $W_{P,ab}$, is updated according to the STDP rule,

$$\Delta W_{P,ab} = W_{P,ab} + \eta \int \rho_a(t)\rho_b(t+s)K_{STDP}(s)ds \tag{7}$$

where $\rho$ represents the spike trains (delta-functions at the time of the spikes), $\eta$ is the learning rate, and $K_{STDP}$ is an asymmetric exponential kernel with time constant $\tau_{STDP}$. The learning rate is set so that a single seizure episode maximally changes synaptic weights by 1/3. The STDP-induced connectivity change is reported as $\Delta W_E$, and the resultant connectivity is analyzed according to its spatial property. Projections from neuron $b$ are divided spatially into two parts: those going to neurons at one side ($W_{E,ab}$ in which $a$) and those going to the other side ($a$). The spatial projection bias of neuron $b$ is defined as the difference between the summation of these two parts, $\sum_{a<b} W_{E,ab} - \sum_{a>b} W_{E,ab}$, summed over the index $a$.

## Local field potential (LFP)-like signal readout in the integrate-and-fire spiking model

LFP-like signal is modeled by the reference weighted sum proxy method (*Mazzoni et al., 2015*). Briefly, LFP is modeled as being proportional to a linear combination of excitatory and synaptic currents, $I_E - \alpha I_I$, where $\alpha = 1.65$ and $I_E$ is temporally delayed by 6 ms, as suggested by Mazzoni et al. Synaptic current contributed by neighboring cells decays exponentially with spatial constant $\sigma_{LFP} = 0.025$ (unit: normalized spatial scale).

## Generalized model of exhaustible inhibition

The generalized model of exhaustible inhibition is modelled analogously to the rate model except for equation 3. Instead of specifically considering transmembrane chloride dynamics, an abstract dynamic variable, $z$, is used to quantify the effectiveness of inhibition: $g_I(x,t) \leftarrow zg_I(x,t)$. The variable $z$ summarizes numerous factors that may contribute to inhibition effectiveness, including chloride gradient, interneuron excitability, short-term plasticity, etc. It is modelled by first-order kinetics

$$\tau_z \frac{\partial z}{\partial t} = z_\infty - z \tag{8}$$

where its steady state, $z_\infty$, depends on the intensity of inhibition usage, $z_\infty = H(g_{I,th} - g_I)$, where $H$ is a Heaviside step function. When inhibition is strong (above $g_{I,th}$), its effectiveness decreases exponentially with time constant $\tau_z$.

## Two-dimensional model

The 2-D rate model is a based on a bounded, circular-shape two-dimensional space. The simulation neural network is generated from a 50 by 50 partitioned square space and then removing the population that falls outside the round boundary (100 by 100 for high video quality in *Figure 1—video 1* and *Figure 1*). Recurrent connections are calculated by two-dimensional Gaussian convolution in a direct generalization of the 1D case. A round space is used to avoid inhomogeneous boundary effects along specific directions.

## Quantifying seizure activity in model and patient recordings

For rate models, the duration and spatial involvement of a model seizure is defined as the convex hull constructed from space-time points at which population neuronal firing rate is more than 10% of the maximal firing rate, $Conv\,((\boldsymbol{x},t):f(\boldsymbol{x},t)>0.1f_{max})$. Successful seizure onset is defined as when the provoked seizure activity ($f>0.1f_{max}$) lasts more than 5 seconds without any requirement of excitatory current input from outside the neural network. Seizures are partitioned into the following stages: pre-ictal, ictal-tonic (after successful seizure onset), ictal-clonic (after emergence of repetitive bursting activity), pre-termination (after disappearance of all tonic-firing regions), and post-ictal (after all neural population firing rates drop below 0.1 $f_{max}$). Transitions between stages are identified visually.

Traveling wave velocities are calculated by least squares linear regression (*Liou et al., 2017*). For patient data, velocity is inferred by regressing multiunit spike timings against their spatial information (significance level = 0.05). For rate models, timings of firing rate peaks are regressed (significance level = $10^{-3}$). Circular standard deviation (20-second window) is used to quantify traveling wave directional variability (*Berens, 2009*).

Spiral waves centers are identified by finding phase singularity points (*Iyer and Gray, 2001*). At a given time $t$, the instantaneous phase, $\psi(\boldsymbol{x}, t)$, of a population is determined by the state of the membrane potential – threshold plane. $\psi(\boldsymbol{x}, t) = angle(V(\boldsymbol{x}, t) - V_m(\boldsymbol{x}) + i(\phi(\boldsymbol{x}, t) - \phi_m(\boldsymbol{x})))$. $V_m(\boldsymbol{x})$ and $\phi_m(\boldsymbol{x})$ are the median voltage and threshold throughout the whole seizure episode at location $\boldsymbol{x}$. Singular points are found by looking for a closed-path, $l$, that circulates the immediately surrounding points, where $\oint_l \nabla \psi \cdot d\boldsymbol{l} = \pm 2\pi$.

# Acknowledgements

This work was supported by the National Institutes of Health through National Institute of Neurological Disorders and Stroke grants R01-NS084142, R01-NS095368, and R01-NS110669, the Gatsby Charitable Trust, the Simons Foundation, and NSF NeuroNex Award DBI-1707398. We thank Steven Siegelbaum, Kenneth Miller, Sean Escola, Ning Qian, and colleagues at Columbia Neurotheory Center for their useful discussions and suggestions.

# Additional information

### Funding

| Funder | Grant reference number | Author |
|---|---|---|
| National Institute of Neurological Disorders and Stroke | R01-NS084142 | Jyun-you Liou<br>Elliot H Smith<br>Catherine Schevon |
| Gatsby Charitable Foundation | | Jyun-you Liou<br>Larry Abbott |
| Simons Foundation | | Jyun-you Liou<br>Larry Abbott |
| National Science Foundation | NeuroNex Award DBI-1707398 | Jyun-you Liou<br>Larry Abbott |
| National Institute of Neurological Disorders and Stroke | R01-NS095368 | Jyun-you Liou<br>Elliot H Smith<br>Catherine Schevon |
| National Institute of Neurological Disorders and Stroke | R01-NS110669 | Jyun-you Liou<br>Elliot H Smith<br>Catherine Schevon |

The funders had no role in study design, data collection and interpretation, or the decision to submit the work for publication.

### Author contributions

Jyun-you Liou, Conceptualization, Data curation, Software, Formal analysis, Investigation, Methodology; Elliot H Smith, Conceptualization, Resources, Data curation, Writing - review and editing; Lisa M Bateman, Resources, Writing - review and editing; Samuel L Bruce, Software, Formal analysis; Guy M McKhann, Robert R Goodman, Resources, Data curation; Ronald G Emerson, Conceptualization, Resources, Data curation; Catherine A Schevon, Conceptualization, Resources, Data curation, Software, Supervision, Funding acquisition, Validation, Investigation, Project administration, Writing - review and editing; LF Abbott, Conceptualization, Resources, Supervision, Funding acquisition, Investigation, Methodology, Project administration, Writing - review and editing

Author ORCIDs
Jyun-you Liou https://orcid.org/0000-0003-4851-3676
Elliot H Smith https://orcid.org/0000-0003-4323-4643
Catherine A Schevon https://orcid.org/0000-0002-4485-7933

Ethics
Human subjects: The informed consent and consent for academic publication were obtained from participants prior to their epilepsy surgeries. The Columbia University Medical Center Institutional Review Board approved the research (protocol number: AAAB-6324).

Decision letter and Author response
Decision letter https://doi.org/10.7554/eLife.50927.sa1
Author response https://doi.org/10.7554/eLife.50927.sa2

## Additional files

### Supplementary files

- Source data 1. Patient A Seizure.
- Source data 2. Patient B Seizure 1.
- Source data 3. Patient B Seizure 2.
- Source data 4. Patient B Seizure 3.
- Transparent reporting form

### Data availability

The human seizure data used in this manuscript is currently restricted from open sharing by university policy, but may be shared with qualified investigators under appropriate institutional protections upon request. Source data files with some anonymised data have been provided. The MATLAB code for the computational model is available at https://github.com/jyunyouliou/LAS-Model (copy archived at https://github.com/elifesciences-publications/LAS-Model).

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
