## [Decision Letter]

**Acceptance summary:**

The computational model proposed by the authors is interesting, scientifically but also clinically since it simulates the mechanisms underlying seizures typically seen in focal cortical dysplasia. We are excited by this paper because the model has the ability to mimic the tonic-clonic transitions and the spatial expansion associated with EEG synchronization typical for these seizures. The authors also provide computational support for their overarching hypothesis that the seizure termination is caused by exhaustion of inhibition.

**Decision letter after peer review:**

Thank you for submitting your article "A theoretical model for focal seizure initiation, propagation, termination, and progression" for consideration by *eLife*. Your article has been reviewed by three peer reviewers, including Jan-Marino Ramirez as the Reviewing Editor and Reviewer #1, and the evaluation has been overseen by Ronald Calabrese as the Senior Editor. The following individuals involved in review of your submission have agreed to reveal their identity: Kevin Staley (Reviewer #2); A. N. Khambhati (Reviewer #3).

The reviewers have discussed the reviews with one another and the Reviewing Editor has drafted this decision to help you prepare a revised submission.

Summary:

In this work, Liou et al. develop and study a biophysical network model that recapitulates a repertoire of seizure dynamics observed during microelectrode recordings from human epileptic cortex. The authors identify biophysically plausible failure points of the otherwise healthy neocortical circuit model by pushing the model system to seizure threshold, via excitatory input, and parametrically varying synaptic conductance dynamics, spike-timing dependent plasticity, and background noise. Liou et al. demonstrate that inhibition and adaptation play critical roles in the spatiotemporal progression of seizures, from onset to termination; and that a naïve network model undergoing a breakthrough seizure can be subject to plasticity rules that increase the probability of subsequent, spontaneous seizures. This study is substantially strengthened by the link between model parameters and naturally occurring seizure dynamics. The idea of using the model as a test-bench to identify putative dysfunctional elements of a patient's seizure network is highly compelling. Strengths: the model generates realistic ictal activity, particularly progression and tonic-clonic transition, that closely conforms to human ictal EEG recordings. The model is less abstract than Epileptor.

Essential revisions:

1) The model does not generate spontaneous seizures, as suggested by the title phrase "model for focal seizure initiation". The model does not intrinsically terminate seizures; termination is not very clearly addressed but appears to involve a global "γ" inhibitory factor that is imposed on the network. Thus the use of "termination" in the title is also not advisable. The model excels at reproducing seizure propagation, the ictal wavefront proposed by the authors, and transitional EEG phenomena observed during seizures such as tonic-clonic transitions. The paper should be written from the title onward to clarify the strengths and limitations of the model, as suggested in the following recommendations:

2) With regards to Seizure initiation: The model is designed to reproduce the spatiotemporal elements of ictal EEG, and is based on the mechanistic hypotheses of the Abbott laboratory. The model does an excellent job of reproducing the ictal EEG and the ictal wavefront, which strengthens the evidence for the underlying mechanistic ictal hypotheses. Seizure propagation is the strength of this model.

a) Without any epochs of normal activity it is difficult to appreciate how well this model reflects the structure and function of human neural networks that do many things and seize only rarely. For example, the activity driven by the white noise inputs is not shown or compared to ictal activity. Does a white noise input applied at one node propagate through the network? If so, what is the nature of that propagating activity?

b) Figure 2—figure supplement 2 characterizes the exogenous input that triggers seizure activity. Without normal activity for comparison, it is difficult to appreciate from Figure 2—figure supplement 2 the nature of this ictogenic input. How does it compare in amplitude / duration to the white noise? If the ictogenic input needs to be sufficient to overcome inhibition, as appears to be the case from Figure 2—figure supplement 2, this would not be normal input, but rather input from an external epileptic focus. In that case, the model is one of seizure propagation, not seizure initiation.

c) Discussion paragraph four suggests seizures are shut off by an exogenous process that does not have a clear neurophysiological correlate (see point 2).

d) If 1a-c are accurate, the model does not generate spontaneous seizures – they are exogenously triggered and terminated. The model should be represented as a propagation model – how seizures spread through normal cortex. The title, Abstract and Discussion should reflect this strength, vs. the interictal-ictal transitions that rely on exogenously applied mechanisms.

3) With regards to Seizure spread: the model does not clarify the relationships of the 3 modulators of inhibition that are employed in this model: ionic gradients (Equation 3), the z factor (Equation 8) and the Î³ factor. Ionic factors are described in detail e.g. pp 10. The z factor is described in much less detail. The Î³ factor is also not described in detail but is suggested to be a global factor that inhibits the entire network equally (Figure 1A and Discussion paragraph four) and is responsible for seizure termination. Termination of seizure activity by a globally imposed external factor is a significant limitation of the model, at least in terms of how the model is currently represented e.g. in the current title. It would be important to meaningfully link a global termination factor such as Î³ to realistic candidate termination processes (e.g. Krishnan and Bazhenov J Neruosci 2011), that can be instantiated in a physiologically feasible manner.

a) The relationships between ionic determinants of inhibition (Equation 3 in the model) and the global inhibition efficacy factor z (Equation 8) and the Î³ factor are not clear.

b) It is not clear how these 3 inhibitory factors contribute to some of the transition phenomena such as those highlighted by the various white symbols in Figure 2A and Figure 2—figure supplement 4.

c) It would be very useful to show more than 1 seizure in the manner of Figure 2A, with the initiating conductance, average GABA conductance, Î³ conductance, and the z inhibitory conductance modulator plotted on the same time scale and amplitude scales.

4) The "Mexican hat" wiring of the neocortex requires more justification than the current references to other modeling studies. Interneurons are so called because they project locally; however in this model the interneurons have the longer range projections and the principal cells have the shorter connections. Rationalizing this connectivity from ictal surround inhibition is a circular argument – the network should start with the most accurate connectivity data available.

Please consider also the following comments:

5) The authors might add a more elaborate discussion about the clinical applications of a model with a more generalized description of biophysical mechanisms. How might the model help predict the therapeutic yield of different forms of therapy for a patient? Or test new therapies?

6) How might the authors reconcile the model's behavior of a gradual seizure termination process, one in which discharges progressively slow and the network becomes desynchronized, with the common observation on clinical iEEG montages of sudden stopping of ictal activity and subsequent suppression/quieting of activity? Please address this question in the Discussion.

7) A major assumption of this study is that any neuronal network, even one that resembles a healthy network, is capable of producing a seizure if provoked with an external input that is of sufficient strength and duration. Perhaps the authors might want to add a discussion on what the initial "external excitatory input" might represent in the context of a real-world breakthrough seizure? Is it possible that certain types of connectivity profiles and topographical distributions of conductances can make a breakthrough, and/or subsequent spontaneous seizure more likely (before any particular re-modelling has occurred)?

8) The authors show that pre-seizure discharges can travel toward the ictal core via centripetal connections formed after STDP remodeling. This raises several interesting thoughts about the putative role of discharges in seizure generation and maintenance, however the phenomenon is not further discussed in the manuscript. Could the authors discuss the functional significance of discharge travelling waves that converge onto the ictal core, rather than in the opposite direction?

9) What, if anything, might the model suggest about mechanisms underlying shorter sub-clinical events and burst-like epileptiform events that are not necessarily considered seizures? Do these events represent edge-cases of the proposed model? The authors may add these considerations into the Discussion.

10) The specificity of the mechanism associated with spiral wave termination is not clear. Did the wave terminate due to a non-specific "global" input or because the globally synchronizing pulse hit one or more of the correct targets to terminate the seizure? Could duration or direction of the pulse relative to the velocity of the wave impact the likelihood that the wave will terminate?

11) It seems that the seizure types studied here only occur in a small sample of the patients (2 patients included / 5 patients excluded). The authors should comment on the specificity of their results on patients with certain forms of epilepsy / types of seizures unique to those included/excluded.

---

## [Author Response]

Essential revisions:1) The model does not generate spontaneous seizures, as suggested by the title phrase "model for focal seizure initiation". The model does not intrinsically terminate seizures; termination is not very clearly addressed but appears to involve a global "γ" inhibitory factor that is imposed on the network. Thus the use of "termination" in the title is also not advisable. The model excels at reproducing seizure propagation, the ictal wavefront proposed by the authors, and transitional EEG phenomena observed during seizures such as tonic-clonic transitions. The paper should be written from the title onward to clarify the strengths and limitations of the model, as suggested in the following recommendations:

We agree that the main model, without any prior remodeling, does not generate spontaneous seizures. Therefore, in the revised title, we replaced 'initiation' with 'onset'. Here, 'onset' is used to encompass the following findings in this manuscript

1) Figure 4 – distribution of recurrent inhibition determines electrographic signatures of seizure onset subtypes.

2) Figure 2—figure supplement 2 – a successful seizure onset requires the triggering stimulus to have adequate intensity, duration, and spatial extent.

3) Figure 2—figure supplement 3 – a successful seizure onset is more likely to be triggered in a network with less robust inhibition and adaptation

Regarding termination, seizures do terminate spontaneously except in the case of spiral wave seizures (Figure 7). By saying the seizure terminates spontaneously, we mean that the seizure terminates due to conductances that are triggered autonomously by network activity. This is in contrast to the scenario shown in Figure 7, in which the seizure is terminated by an 'exogenous' pulse, whose activation is not dependent on the network's activity itself.

The progressive build-up of spatially non-specific recurrent inhibition is the reason that model seizures can terminate spontaneously. The level of this spatially diffuse recurrent inhibition depends on the neural network's activity in that neurons in the network need to fire to activate recurrent inhibition. We interpreted conductance changes that are totally dependent on the network's activity itself to be intrinsic and spontaneous. Please also see our response to major comments 2c and 3 for further discussion about the nature of this global recurrent inhibition.

That being said, we do agree and appreciate the reviewers' suggestion to remove the phrase 'termination' from the title. Indeed, we showed that termination is just one of the possible endpoints of the model dynamics (Figure 6 to 7). We therefore substitute 'termination' with a more generalized phrase 'evolution' to encompass both endpoints (termination or spiral seizures). Therefore, the newly revised title is: A model for focal seizure onset, propagation, evolution, and progression.

2) With regards to Seizure initiation: The model is designed to reproduce the spatiotemporal elements of ictal EEG, and is based on the mechanistic hypotheses of the Abbott laboratory. The model does an excellent job of reproducing the ictal EEG and the ictal wavefront, which strengthens the evidence for the underlying mechanistic ictal hypotheses. Seizure propagation is the strength of this model.a) Without any epochs of normal activity it is difficult to appreciate how well this model reflects the structure and function of human neural networks that do many things and seize only rarely. For example, the activity driven by the white noise inputs is not shown or compared to ictal activity. Does a white noise input applied at one node propagate through the network? If so, what is the nature of that propagating activity?

Figure 2—figure supplement 1 has now been greatly expanded to provide comparison between baseline activity (Panel B and C) versus seizures (Panel D and E).

Panel B is a raster plot of baseline activity of a 1D spiking model before application of any seizure provoking inputs. With threshold noise, the network maintains a chaotic, non-trivial neuronal firing pattern indefinitely (approximately 0.74 Hz). Panel C shows the LFP-like signal readouts. Colors indicate where the signals are calculated in Panel B. The LFP-like signals look like a normal, active EEG background. Panel D and E show the same network's activity after seizure onset (ictal clonic stage). Clear difference can be seen in both raster plots and LFP-like signal readouts.

We chose to the spiking model to compare baseline activity versus seizures because only spiking models have LFP-like signal readouts (Panel C and E).

Regarding the propagation of white noise, any input given to the network is low-pass filtered because of membrane dynamics and then projected recurrently to produce an image of local excitation with surround inhibition. A snapshot of membrane potential across the network during baseline activity is further shown here.

**Author response image 1. respfig1:** You can see several neighborhoods of depolarization caused by recently spiked neurons, which can be identified by their hyperpolarized membrane potentials (remember the spiking rule is: V ← V-20mV).

In brief, the network exhibits a stable, healthy state at baseline. It is the external inputs that trigger the model seizures. This has been re-emphasized in the revised manuscript:

Results:

"In its original form, the model does not spontaneously seize. Instead model seizures are initiated by transient focal excitatory input. This model can be considered a model of a healthy neural circuit, with the degree of resistance to seizure controlled by its parameters."

Discussion:

"Without modification by STDP, the network is reminiscent of a healthy, non-epileptic brain – it maintains a non-trivial baseline firing rate, does not spontaneously seize, but can be provoked into a full-blown seizure."

b) Figure 2—figure supplement 2 characterizes the exogenous input that triggers seizure activity. Without normal activity for comparison, it is difficult to appreciate from Figure 2—figure supplement 2 the nature of this ictogenic input. How does it compare in amplitude / duration to the white noise? If the ictogenic input needs to be sufficient to overcome inhibition, as appears to be the case from Figure 2—figure supplement 2, this would not be normal input, but rather input from an external epileptic focus. In that case, the model is one of seizure propagation, not seizure initiation.

As our response to comment 2a, we agree that the model does not seize spontaneously before remodeling, and we have provided normal activity for comparison. Therefore, as our response to question 1, we have taken off 'initiation' from our title.

Figure 2—figure supplement 2 has been revised to provide more straightforward understanding of the ictogenic inputs.

Panel A shows a few examples of the external inputs. The green boxes indicate the spatial extent and duration of the inputs (for example, the left panel: 1-second long, applied to neurons located from 0.475 to 0.525). The inputs' intensity is also reported (100 pA) both in the figure and its associated legend. You can see the input in the right panel successfully triggered a seizure, but the inputs in the two panels to the left do not, although some after-discharges are seen in the middle panel. Panel B summarizes the network's response to a variety of ictogenic input parameters.

In contrast to the ictogenic inputs, the noise we applied to the models was not restricted to a specific location or duration. Background current noise (Is) is applied to the rate model shown in Figure 6 and 7. Its strength, quantified by its diffusion coefficient, not amplitude, is reported in the figure legend (200 pA^2^/ms). The spiking model also has built-in threshold noise, which means there is no definite spiking threshold. Instead, the probability of spiking is a sigmoid function of each neuron's membrane potential. Threshold noise was also present in all neurons at all times in any spiking model.

We have revised our Discussion to discuss the nature of the ictogenic input.

"... In this sense, perhaps the most straightforward real-world interpretation of the external stimuli in our simulations is the focal electrical zap given during intra-operative cortical mapping (Ritaccio et al., 2018). The suprathreshold stimuli are therefore analogous to the electrical shock given during convulsive therapies (Spellman et al., 2009). More generally, the stimuli are qualitatively equivalent to any factor that cause a neighborhood of neurons to depolarize, such as anoxic depolarization caused by local ischemia (Somjen, 2001) or discharges triggered by transcranial magnetic stimulation (Lisanby et al., 2003)."

It is important to clarify this ictogenic input is not necessarily from an external epileptic focus. In such conditions, the 'source' epileptic focus would have to remain outside the model cortex yet still produce strong depolarizations in a spatially restricted area of the model cortex. Future work will focus on the mechanisms of spontaneous seizure onset with various subtypes of physiological ictogenic inputs to the model cortex.

c) Discussion paragraph four suggests seizures are shut off by an exogenous process that does not have a clear neurophysiological correlate (see point 2).

We cannot find any text suggesting the seizures are shut off by an exogenous process. Also, we would like to clarify again that the seizure terminates by itself. No external input is required to terminate the seizure other than the status epilepticus scenario (Figure 7).

Please see our response to comment 1 and 3 for detailed discussion about spontaneous seizure termination.

Regarding possible neurophysiological correlates of the spatially non-specific recurrent inhibition, which leads to seizure termination in our model, please see:

"... Anatomically, such widespread inhibition may be mediated by long-distance, cross-areal projections that preferentially project to inhibitory interneurons in their target circuits (Zhang et al., 2014; Sun et al., 2019). In addition, focal seizures may depress brain-wide activities via disrupting subcortical structures (Feng et al., 2017) and ascending activation systems (Blumenfeld, 2012). Large-scale non-neural mechanisms that are not specifically modelled in our study may also contribute to seizure termination. For example, global hypoxia secondary to seizure-induced cardiopulmonary compromise may activate ATP-sensitive potassium channels in pyramidal neurons in a spatially non-specific way (Ching et al., 2012), thereby terminating seizures by building resistance ahead of the ictal wavefronts and generating termination dynamics, analogous to the global recurrent inhibition effect modelled in our study..."

d) If 1a-c are accurate, the model does not generate spontaneous seizures – they are exogenously triggered and terminated. The model should be represented as a propagation model – how seizures spread through normal cortex. The title, Abstract and Discussion should reflect this strength, vs. the interictal-ictal transitions that rely on exogenously applied mechanisms.

To summarize our response to a-c, we agree this is not a model showing how the seizure mechanism initiates – they are triggered by external inputs. However, the seizure does spontaneously terminate as recurrent global inhibition is activated by the network's activity itself. The title has been revised (see response to Comment 1). All the relevant revisions have been presented in our responses in the previous comments (also to Comment 7 and 9). We appreciate the opportunity to clarify these issues in the manuscript as aforementioned.

3) With regards to Seizure spread: the model does not clarify the relationships of the 3 modulators of inhibition that are employed in this model: ionic gradients (Equation 3), the z factor (Equation 8) and the Î³ factor. Ionic factors are described in detail e.g. pp 10. The z factor is described in much less detail. The Î³ factor is also not described in detail but is suggested to be a global factor that inhibits the entire network equally (Figure 1A and Discussion paragraph four) and is responsible for seizure termination. Termination of seizure activity by a globally imposed external factor is a significant limitation of the model, at least in terms of how the model is currently represented e.g. in the current title. It would be important to meaningfully link a global termination factor such as Î³ to realistic candidate termination processes (e.g. Krishnan and Bazhenov J Neruosci 2011), that can be instantiated in a physiologically feasible manner.a) The relationships between ionic determinants of inhibition (Equation 3 in the model) and the global inhibition efficacy factor z (Equation 8) and the Î³ factor are not clear.

We apologize that the equations were not explained clearly enough and may have caused some misunderstandings.

Chloride dynamics and variable z We want to ensure that 'z', being called by the reviewers a 'global' inhibition efficacy factor, is not misinterpreted as 'global' in a spatial sense. It is a local variable, controlling the strength of inhibition conductance onto excitatory cells. In other words, z is a function of x and t. A simple way to understand z, is that it is a general (i.e. non-biophysically based) factor that represents each inhibitory neuron's ability to convert excitatory inputs into inhibitory outputs which project to its partner excitatory neurons.

In this manuscript, the cell membrane potential dynamics were either described by Equation 1+2+3+4 (the main model) or 1+2+8+4 (generalized model of exhaustible inhibition), and the latter model (z) was only used once, presented in Figure 2—figure supplement 4. There is no model presented here in which z and chloride dynamics exist at the same time.

Instead, Equation 8 is a general way to describe inhibition strength. Instead of limiting our exploration by saying that the chloride gradient is the only mechanism that causes exhaustible local inhibition strength, we acknowledge that there may be multiple factors that control local inhibition strength and they may be exhausted by repetitive use. Therefore, we chose a simple formula (Equation 8) to describe our hypothesis: a general model of exhaustible inhibition. Equation 8 simply says that heavy use of local inhibition (>gI.thr) can decreases the strength of z. The reason why we developed this generalized inhibition exhaustion model (Figure 2—figure supplement 4) is to ensure our readers understand the idea that 'inhibition exhaustion' causes seizure territory expansion, and chloride dynamics represent one of the better candidate mechanisms underpinning such inhibition exhaustion. A simple phenomenological way to describe local inhibition strength, Equation 8, in replacement of Equation 3, can also generate qualitatively similar dynamics, confirming that the essential focal seizure dynamics studied here are due to exhaustible inhibition.

This message is more clearly described in the revised Discussion:

"The biophysical processes proposed here are certainly not the only mechanisms causing seizure evolution. Instead, they should be considered as representative candidates, with the suggestion that other mechanisms should share the features of the ones we have proposed. Processes that produce similar effects on network dynamics and that operate over the same time scales may also contribute to the generation of seizure dynamics, as exemplified by our phenomenological model, in which the general variable z controls local effectiveness of inhibition rather than the transmembrane chloride gradient."

Parameter γ: γ is a parameter that describes how much percentage of projections from excitatory cells to inhibitory cells is distributed in a spatially non-specific way. γ is the 'global' factor, not z. γ is a constant regardless time or space. γ is fixed during simulations.

We would like clarify that the recurrent inhibition pathway, γ, in our model is not an external factor. According to Equation 5, recurrent inhibition, gI(x,t), is calculated by the following 2 steps:

First, calculating the recurrent inhibition projections (PI):PI(x,t)=(1−γ)(A(x,t)*N(0,σI2)+γ∫A(x,t)dx

Then, PI(x,t) is temporally filtered with an exponential kernel, SI, and scaled with gI¯gI(x,t)=gI¯(PI*SI)

The activation of the spatially non-specific pathway, namely, the 2^nd^ term of the upper equation, is dependent on the model network's activity itself. The equation indicates that network's activity is spatially integrated and fed back into the network. This is in contrast to the seizure initiation stimulus (Iext), which is not dependent on the network's activity at all.

Please notice that we do not call this 'intrinsic' in an anatomical sense. The real-world interpretation of the global recurrent inhibition should not be limited to cortico-cortical projections. They may represent multi-synaptic projections mediated by cortex-deep structure-cortex loops (e.g. cortico-striato-thalamic loop). They may even represent non-synaptic mechanisms that are activated by cortical activity. The γ (spatially non-specific) term should be interpreted as a simplified collection of all these large-scale effects. For further discussion, please see our response to comment 1 and 2c.

b) It is not clear how these 3 inhibitory factors contribute to some of the transition phenomena such as those highlighted by the various white symbols in Figure 2A and Figure 2—figure supplement 4.

First, we changed the symbols from white to green because we received some readers' feedbacks that white symbols may be mistaken as maximal firing in the spatiotemporal diagrams.

Chloride dynamics and variable z: Equation 3 and 8 are qualitatively interchangeable and both focus on inhibition robustness (see our response to comment 3a). Since Equation 3 (chloride dynamics) is the main model, we will use simulation results that use Equation 3 to clarify this point.

Inhibition robustness plays a major role in controlling seizure propagation (see Figure 2D and Figure 2—figure supplement 3). The stronger it is, the more difficult for the ictal wavefront to be established and move forward. This has implications in the following transition phenomena marked in Figure 2A and Figure 2—figure supplement 4.

1) The success of seizure onset, marked with the green diamond signs (transition from the pre-ictal to ictal-tonic stage).

Robust inhibition prohibits establishment of a tonic-firing area. As shown in Figure 2F, notice that the lower τCl is (fast chloride clearance = robust inhibition), the less likely a seizure can be initiated.

2) The annihilation of the ictal wavefront, marked with the green arrows (transition from the ictal-clonic to pre-termination stage)

As shown in Figure 2C, the ictal wavefront's movement is determined by inhibition robustness and adaptation current adaptation. A robust inhibition slows down the expansion of the ictal wavefront (Figure 2D). Because of slow expansion, the wavefront is caught up by the sAHP-mediated transition early. Thus, the ictal wavefront is annihilated early and the overall seizure territory is small (Figure 2D).

3) Seizure termination, marked with the green triangles (transition from the pre-termination to post-ictal stage)

Because the ictal wavefront is annihilated early, seizures that are initiated in a network model with more robust inhibition also end early.

Parameter γ: We found that the distribution of recurrent inhibition plays a major role in seizure onset patterns. Figure 4 is devoted to illustrate this finding. Also, high γ means local recurrent inhibition is weak, that is more distributed to the global side. Therefore, it is unsurprising that less strong focal stimuli are required for a successful seizure onset. Given its straightforwardness, this observation is not particularly described in our manuscript.

Regarding the paper provided by the reviewers (Krishnan and Bazhenov J Neruosci 2011), there is no concept of space at all in their model. The termination process we proposed here is very different from theirs and not necessarily ionic in nature. A summary of seizure termination mechanism in our model is included in Discussion:

"In our model, spontaneous seizure termination is caused by progressive build-up of inhibition, which is used to model the widespread effect of focal seizures (Burman and Parrish, 2018; Liou et al., 2019)."

Our seizures are terminated because the seizure territory is large enough to create adequate global recurrent inhibition, which curbs further propagation of the ictal wavefront and then seizure termination. This is very different from the Krishnan et al.'s model, which is totally dependent on ionic dynamics. Therefore, we opt not to refer to this study.

c) It would be very useful to show more than 1 seizure in the manner of Figure 2A, with the initiating conductance, average GABA conductance, Î³ conductance, and the z inhibitory conductance modulator plotted on the same time scale and amplitude scales.

We have greatly expanded Figure 2—figure supplement 1 (the corresponding spiking model to Figure 2A) to provide one more in detail seizure analysis in a style similar to Figure 2A.

We would also like to clarify some terminology used in the reviewers' comments. We apologize our original manuscript might have been not clear enough so some misunderstandings might have happened.

1) "Initiating conductance" – Instead of conductance, the model seizures are actually triggered by direct current injections (the deterministic term of Iext: Id). The spatiotemporal extents have been marked in Figure 2A (the green box), Figure 2—figure supplement 1 (red-shaded area), Figure 2—figure supplement 4 (the green box). The amplitudes are reported in the legends.

2) "Average" GABA conductance and Î³ conductance.

Because GABA conductance is a variable of x and t, we believe the reviewer is probably asking for average GABA conductance in terms of time or space. However, we are afraid that averaging GABA conductance either over time or space might not provide insights what drive evolution of seizure dynamics because the effect of GABA conductance is local (specific to each neuron) and time-sensitive (τ=15ms). Instead, we chose to show the spatiotemporal diagram of [Clin], which controls the effectiveness of GABA neurotransmission and ictal wavefront propagation. Information regarding [Clin] now has been shown not only in Figure 2C but also in Figure 2—figure supplement 1A.

Regarding 'γ conductance', we apologize again for this misunderstanding, but γ is not a conductance. Instead, it is a percentage, a parameter fixed throughout simulations. Please also refer to our response to comment a.

Regarding the z inhibition conductance modulator, we have also expanded Figure 2—figure supplement 4, the generalized inhibition effectiveness model, in a similar style similar to Figure 2A-C.

4) The "Mexican hat" wiring of the neocortex requires more justification than the current references to other modeling studies. Interneurons are so called because they project locally; however in this model the interneurons have the longer range projections and the principal cells have the shorter connections. Rationalizing this connectivity from ictal surround inhibition is a circular argument – the network should start with the most accurate connectivity data available.

Long-range projecting interneurons are not required to establish "Mexican-hat" connectivity. Notice that recurrent inhibition in this model is actually the result of di-synaptic projections (E→I and then I→E, with interneurons reacting instantaneously and linearly). However, recurrent excitation is only mediated by 1 synapse (E→E). The two terms: σE→I and σI→E contribute to σI. In contrast, σE is only contributed by σE→E. In our study, neuronal synaptic projections are spatially Gaussian and mutually independent; therefore, σI2=σE→I2+σI→E2.

Therefore, the Mexican-hat connectivity simply says σE→I2+σI→E2>σE→E2. It does not specify the existence of any long-range projecting interneuron. Furthermore, σE→I2+σI→E2>σE→E2 does not require that σI→E>σE→E if σE→I is sufficiently large. The most extreme scenario, which is shown in Figure 1A, is that all interneuron projections are strictly local. They do not even send out projections to their neighboring columns except their own partner pyramidal neurons (Figure 1A, blue projections). Only excitatory cells send out projections to other columns (Figure 1A, red projections). In other words, the cartoon in Figure 1A shows a scenario of σE→I>σE→E. Therefore, σI>σE is guaranteed.

We agree that whether σE→I2+σI→E2>σE→E2 is true in the neocortex is debatable. In our opinion, the question if σE→I2+σI→E2>σE→E2 is true anatomically is ill-defined, because projection patterns of cortical principle neurons vary significantly according to cell subtypes and cortical layers (Harris and Shepherd, 2015). However, even if there is no significant statistical difference between σE→I and σE→E, because σI→E is not 0 (Karnani et al., 2014), it is not unreasonable to expect σE→I2+σI→E2>σE→E2, as it involves the combination of two projections rather than just one.

We agree with the reviewers that the Mexican hat connectivity assumption in our study should be more appropriately called a hypothesis, because anatomical evidence is not currently available and may not even be possible. However, we would like to provide three reasons why this may be a reasonable hypothesis at the neocortex.

First, theoretically, Mexican hat connectivity has been widely adopted in many models, including visual cortex orientation selectivity (Kang et al., 2003), grid cell pattern formation in the entorhinal cortex (Burak and Fiete, 2009), working memory in the frontal cortex (Wang, 2001), and multiplicative neural responses in the parietal cortex (Salinas and Abbott, 1996). Although distances between neurons in these models are defined functionally, not physically, the success of Mexican hat connectivity in multiple cortical areas may reveal a common wiring strategy of the neocortex.

Second, we hold a different opinion regarding the implication of the ictal surround inhibition observation originally reported by Prince and Wilder (Prince and Wilder, 1967). Prince and Wilder applied penicillin to induce focal epileptic discharges. They found that the post-synaptic potentials (PSPs) caused by 'individual' epileptic discharges transition from net excitatory to net inhibitory as the distance of intracellular recordings from the seizure focus increases. If we agree that the sharp epileptic discharges can be seen as a biological approximate of a Dirac pulse, then, their PSPs can be considered as the impulse response function, which is Mexican-hat shaped. Therefore, we humbly ask the reviewers to agree with us that citing this experiment is not a circular argument.

Third, focal cortical electrical pulse stimulation shows a spatially more extensive distribution of inhibitory responses than excitatory ones (Butovas and Schwarz, 2003) (Please see the reference's Figure 3 and Figure 6). In our model, this is similar to give a pulse to one neuronal population, asking what are the spatial distributions of the effects caused by the pulse. The Mexican connectivity in our model captures this experimentally observed surround inhibition response.

The Results has been revised to clarify this issue and refer the readers for more information regarding interneuron simplification in Material and Methods.

Results:

"Model neurons are recurrently connected by direct excitatory projections (Figure 1A). They also inhibit each other indirectly through di-synaptic pathways via interneurons, whose dynamics are simplified in this study (see Materials and methods for more information regarding interneuron simplification). The effects of recurrent projections between model neurons are hypothesized to be distance-dependent, with the range of di-synaptic recurrent inhibition longer than the mono-synaptic excitatory range, thereby making the spatial distribution of the effective synaptic weights from a model neuron follow a "Mexican hat" structure (Prince and Wilder, 1967; Coombes, 2005; Bressloff, 2014)."

Materials and methods section regarding interneurons and connectivity:

"Inhibitory interneurons are simplified in this study. Their membrane potential dynamics is not specifically modeled, and they react instantly with a monotonic dependence on their synaptic inputs and only project to their corresponding excitatory neurons at the same location. With the help of these instantly reacting interneurons, model neurons are computationally equivalent to emitting both excitatory and inhibitory synaptic projections to each other. Notice that there is no long-range projecting interneurons required (Figure 1A)."

Please consider also the following comments:5) The authors might add a more elaborate discussion about the clinical applications of a model with a more generalized description of biophysical mechanisms. How might the model help predict the therapeutic yield of different forms of therapy for a patient? Or test new therapies?

The Discussion now includes potential therapeutic applications of this model (Please also see our response to comment 8.)

Discussion:

"... This model prediction has potential implications in clinical seizure management. If patterns of interictal discharge traveling waves could be used to reduce uncertainty about the location of subsequent seizures, they could be used as a valuable seizure prediction and diagnostic tool. Furthermore, preventing or reversing traveling wave-induced spatial rewiring may break the pathological, self-enhancing loop, eventually leading to slowdown or reversal of seizure progression."

Regarding a model with a more generalized description of biophysical mechanisms and its clinical applications, also see our revision:

"The biophysical processes proposed here are certainly not the only mechanisms causing seizure evolution. [...] Developing therapies targeting at these mechanisms could therefore achieve broad spectrum anti-seizure effects."

The key information we would like to convey is that therapy should be developed to target mechanisms that underpin the key dynamics, as this would be maximally effective against a variety of focal seizures. Therefore, we can conclude from the mode that therapeutics enhancing inhibition robustness or speeding up adaptation should be encouraged.

6) How might the authors reconcile the model's behavior of a gradual seizure termination process, one in which discharges progressively slow and the network becomes desynchronized, with the common observation on clinical iEEG montages of sudden stopping of ictal activity and subsequent suppression/quieting of activity? Please address this question in the Discussion.

Gradual slowing-down and sudden stopping of ictal activity are not contradictory (Smith et al., 2016). They describe two phenomena that happen at two different time scales. Please see the revised Figure 2—figure supplement 1, Panel F and G.

The seizure stayed in the pre-termination stage for several seconds to slow down. However, this seizure still suddenly stopped from the EEG perspective (Panel G: LFP readouts at the locations marked with blue, brick, and yellow triangles in Panel B).

The subsequent suppression/quieting of activity is quantified in H. Average neuronal firing rates in the seizure territory is significantly less during the post-ictal (blue) than pre-ictal periods (red) (5-second average). For detailed statistics, please see the figure legend.

We chose to illustrate these points in the spiking model because the reviewers asked about iEEG phenomenon. There is no LFP readouts in the rate models. However, the concept is the similar. The sudden stop is a manifestation of the traveling wave dynamics. The last traveling wave emitted electrical activity that manifests as a sudden stop across the seizure territory.

7) A major assumption of this study is that any neuronal network, even one that resembles a healthy network, is capable of producing a seizure if provoked with an external input that is of sufficient strength and duration. Perhaps the authors might want to add a discussion on what the initial "external excitatory input" might represent in the context of a real-world breakthrough seizure? Is it possible that certain types of connectivity profiles and topographical distributions of conductances can make a breakthrough, and/or subsequent spontaneous seizure more likely (before any particular re-modelling has occurred)?

As the reviewers point out, the naÃ¯ve network, before any particular re-modeling, is actually healthy cortical network alike – it does not seize spontaneously but can be provoked to seize if the stimulus is strong enough.

Clinical examples that can be considered the exact replica of the focal excitatory external inputs that trigger focal seizures in our model include:

1) FEAST (focal electrically administered seizure therapy), a novel form of ECT (electroconvulsive therapy), which causes spreading seizures in patients with no known structural brain pathology or history of epilepsy for therapeutic purposes (Spellman et al., 2009).

2) TMS (transcranial magnetic stimulation), another focally applied stimulation modality, which is well known to trigger clinically evident seizures in normal subjects (Schrader et al., 2004). In fact, the mechanism of MST (magnetic seizure therapy), an alternative way than ECT to induce therapeutic seizures, relies on the electrical currents induced by strong, locally applied magnetic fields (Lisanby et al., 2003)

However, we would like clarfy that actually any factor that causes regional depolarization can be the real-world counterparts of this seizure-provoking excitatory inputs. One simple example is anoxic depolarization caused by neuronal hypoxia.

The Discussion in our manuscript has been revised accordingly:

"Without modification by STDP, the network is reminiscent of a healthy, non-epileptic brain – it maintains a non-trivial baseline firing rate, does not spontaneously seize, but can be provoked into a full-blown seizure. [...] Accordingly, responses to the external stimuli, in return, may reveal the network's intrinsic propensity to seize."

Connectivity profiles and topographical distributions of conductances definitely affect seizure susceptibility. One clear example is the centripetal connectivity. In contrast to the naïve network, the network with centripetal connectivity is epileptic: it does not need a net positive input to seize. Although in this study the centripetal connectivity is secondary to seizure-induced synaptic plasticity, it does not mean seizures are the only possible way to cause this.

We, however, did not explore networks with pre-existing spatially inhomogeneous distribution of connectivity or cell properties. Our main goal is to illustrate how the key seizure dynamics emerge from neurophysiological principles instead of studying factors that control seizure thresholds. Future work will involve exploring these ideas for understanding the mechanisms of seizure onset.

For revision regarding centripetal connectivity and seizure susceptibility, please also see our answer to Comment 8.

8) The authors show that pre-seizure discharges can travel toward the ictal core via centripetal connections formed after STDP remodeling. This raises several interesting thoughts about the putative role of discharges in seizure generation and maintenance, however the phenomenon is not further discussed in the manuscript. Could the authors discuss the functional significance of discharge travelling waves that converge onto the ictal core, rather than in the opposite direction?

We agree that this is a fascinating prediction of the model. We published a paper this year reporting similar dynamics in a rodent model, which is now cited in the Discussion, and we are currently working on empirical validation of this model prediction using weeks' worth of interictal human microelectrode array recordings. Therefore, the paragraph discussing plasticity and centripetal connectivity has been completely re-written:

"The model predicts that physiological plasticity mechanisms (STDP in this model) can be hijacked by pathological seizure dynamics (Mehta et al., 1993). [...] Furthermore, preventing or reversing traveling wave-induced spatial rewiring may break the pathological, self-enhancing loop, eventually leading to slowdown or reversal of seizure progression."

9) What, if anything, might the model suggest about mechanisms underlying shorter sub-clinical events and burst-like epileptiform events that are not necessarily considered seizures? Do these events represent edge-cases of the proposed model? The authors may add these considerations into the Discussion.

In our model, short sub-clinical events with burst-like epileptiform discharges can be induced by near-threshold triggers. The examples have now added into Results:

"The sequence of seizure stages is not affected by duration, spatial extent, or intensity of the external seizure-provoking inputs as long as they are adequate to trigger seizure onsets (Figure 2—figure supplement 2). Inputs that are inadequate to initiate a seizure, yet are near enough to the threshold enough for seizure induction, may trigger short-runs of "after-discharges" (Figure 2—figure supplement 2A)."

The series of transitional activities are actually commonly seen during intra-operative cortical mapping. The Discussion has therefore been revised as follows:

"... Varying the external stimulus' strength produces a range of responses from post-stimulation depression, short-run afterdischarges, to full-blown seizures. In this sense, perhaps the most straightforward real-world interpretation of the external stimuli in our simulations is the focal electrical zap given during intra-operative cortical mapping (Ritaccio et al., 2018)... Accordingly, responses to the external stimuli, in return, may reveal the network's intrinsic propensity to seize."

10) The specificity of the mechanism associated with spiral wave termination is not clear. Did the wave terminate due to a non-specific "global" input or because the globally synchronizing pulse hit one or more of the correct targets to terminate the seizure? Could duration or direction of the pulse relative to the velocity of the wave impact the likelihood that the wave will terminate?

A global synchronizing pulse needs to have adequate duration and amplitude to terminate spiral waves in our model. As the reviewers request, the study of pulse duration and result now has been incorporated into the newly revised Figure 7.

In terms of whether there are one or more of the correct targets to terminate seizures, we do not find a consistent location for local pulses to more reliably terminate spiral wave seizures. Stimulating the spiral wave centers is not more effective, as new spiral wave centers tend to emerge at the edge of the region where the pulse is delivered. Instead, we think the reason that global pulse works is because it increases every neuron's firing threshold (Equation 2, spike adaptation). Temporarily, there are not enough neurons available to sustain spiral waves. Thus, the seizure stops.

We opt not to include local pulse study in this manuscript because 1) there is no consistent result 2) we want to emphasize is that there are 2 endpoints that a seizure can stochastically evolve into even all parameters stay the same and 3) there is a pathway that one endpoint can be switch to the other.

11) It seems that the seizure types studied here only occur in a small sample of the patients (2 patients included / 5 patients excluded). The authors should comment on the specificity of their results on patients with certain forms of epilepsy / types of seizures unique to those included/excluded.

Since the model focuses on the physiology of ictal wavefronts and their associated traveling wave dynamics, the patient recordings needed to provide validation necessarily had to demonstrate a clearly evident ictal wavefront. This was true of only 2 of the cases that were available at the time when this work was done. The other cases' microelectrodes, we believe, were positioned outside the seizing brain area (Schevon et al., *2012*; Merricks et al., 2015; Merricks et al., 2020)*–* as the neuronal firings were

1) Not synchronized

2) Not phase-locked with respect to LFPs

3) Not achieving sufficiently high firing rates to cause changes in action potential waveforms

We are aware that whether an ictal wavefront is a universal feature of focal seizures is still controversial, and that this will require an ongoing effort to study this question in both human and animal recordings. However, we would also like to clarify that our model is not totally dependent on the validation provided by the two Utah array recordings. Actually, the Utah data are only required for corroboration of the ictal wavefronts and the associated fast traveling wave velocities, which have been published previously. Other key dynamics of our model seizures, such as tonic-to-clonic transitions, pre-termination slowing, even the traveling wave nature of ictal discharges (Emerson et al., 1995) have long been described. The network's response to the external stimuli also has clear clinical associate, such as responses to cortical mapping. Thus, we hypothesize that the effects described in this model are, in fact, relevant to most if not all types of focal seizures. We note that sampling limitations, which apply to human and animal studies alike, have made this question difficult to address, and has often resulted in significant confusion in the literature. Therefore, we adopt the reviewers' suggestion, adding a devoted paragraph in Discussion addressing this issue:

"Although microelectrode recording plays an important role in validating our model, the key dynamics which are commonly seen behaviorally and observed in macroelectrode recordings, in our opinion, provides an equal, if not more important, support the generality of this model. The question therefore arises why relatively few human microelectrode recordings have demonstrated the existence of ictal wavefronts. Our model, indeed, provides a straightforward explanation – an array not only needs to be positioned at a region that is recruited into the seizure territory but also needs to be close enough to the onset spot. Otherwise, the seizure could have evolved into its pre-termination stage, during which the seizure activity is still slowly propagating but the wavefront has been annihilated."